# Federated Transformer: Multi-Party Vertical Federated Learning on Practical Fuzzily Linked Data

**Zhaomin Wu, Junyi Hou, Yiqun Diao, Bingsheng He**
National University of Singapore, Singapore
zhaomin@nus.edu.sg, {junyi.h,yiqun,hebs}@comp.nus.edu.sg

## Abstract

Federated Learning (FL) is an evolving paradigm that enables multiple parties to collaboratively train models without sharing raw data. Among its variants, Vertical Federated Learning (VFL) is particularly relevant in real-world, cross-organizational collaborations, where distinct features of a shared instance group are contributed by different parties. In these scenarios, parties are often linked using fuzzy identifiers, leading to a common practice termed as *multi-party fuzzy VFL*. Existing models generally address either multi-party VFL or fuzzy VFL between two parties. Extending these models to practical multi-party fuzzy VFL typically results in significant performance degradation and increased costs for maintaining privacy. To overcome these limitations, we introduce the *Federated Transformer (FeT)*, a novel framework that supports multi-party VFL with fuzzy identifiers. FeT innovatively encodes these identifiers into data representations and employs a transformer architecture distributed across different parties, incorporating three new techniques to enhance performance. Furthermore, we have developed a multi-party privacy framework for VFL that integrates differential privacy with secure multi-party computation, effectively protecting local representations while minimizing associated utility costs. Our experiments demonstrate that the FeT surpasses the baseline models by up to 46% in terms of accuracy when scaled to 50 parties. Additionally, in two-party fuzzy VFL settings, FeT also shows improved performance and privacy over cutting-edge VFL models.

## 1 Introduction

Federated Learning (FL) is a learning paradigm that enables multiple parties to collaboratively train a model while preserving the privacy of their local data [27]. Among its various forms, Vertical Federated Learning (VFL) [53] is particularly prevalent form in real-world applications as highlighted in a recent technical report [48]. In VFL, participants possess different features of the same set of instances, where common features, such as names or addresses, serve as *identifiers* (a.k.a. *keys*) to link datasets across these parties.

Real-world applications often necessitate *multi-party fuzzy VFL*, characterized by two key attributes. First, it supports collaboration among *multiple parties*, commonly observed in collaborations across hospitals [33], sensors [52], and financial institutions [38]. Second, it accommodates scenarios where these parties are linked using fuzzy identifiers, such as addresses. Such scenarios are prevalent in applications, as illustrated in an analysis [50] of the German Record Linkage Center [11]. For instance, multiple vehicle rental companies that are fuzzily linked by source and destination addresses in the same city can collaborate to predict travel times.

To illustrate the significance of multi-party fuzzy VFL, consider the application of travel cost prediction in a city through collaboration among taxi, car, bike, and bus companies, as shown in Figure 1. Since personal travel information is private and cannot be shared, VFL is essential.

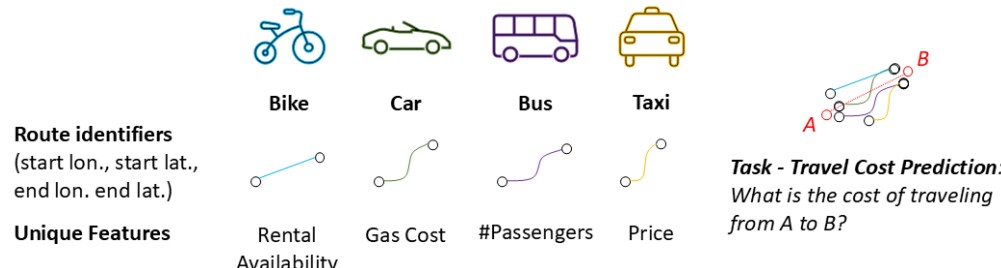

Figure 1: Real application of multi-party fuzzy VFL: travel cost prediction in a city

Additionally, route identifiers - starting and ending GPS locations - can only be linked using fuzzy methods. However, linking closely related source and destination points with multi-party fuzzy VFL can significantly enhance prediction accuracy.

Existing VFL approaches generally address either the multi-party aspect or the fuzzy identifier issue. Several methods [33, 52, 38, 21, 26] facilitate multi-party VFL using Private Set Intersection (PSI) [10] to link datasets. These methods often presume the existence of precise, universal keys, which are not feasible in common VFL scenarios involving fuzzy identifiers. Conversely, other studies [17, 34, 50] propose two-party fuzzy VFL models that utilize cross-party key similarities for training. However, when extended to multi-party fuzzy VFL, these similarity-based approaches encounter significant challenges in performance and privacy. While some methods achieve reasonable performance, they often compromise privacy or incur prohibitive costs.

Despite the potential of multi-party fuzzy VFL, several significant challenges must be addressed for effective implementation. First, as the number of parties with *fuzzy identifiers* increases, maintaining performance becomes increasingly challenging. The addition of parties leads to a quadratic growth in the number of key pairs, an increase in incorrect linkages between fuzzy identifiers, and larger model sizes. These factors collectively heighten the risk of overfitting and adversely impact model performance. Second, the rising costs of preserving privacy intensify as more parties with *correlated data* participate, leading to either significant computational costs [33, 52, 38] or accuracy loss [47]. Third, in a collaboration of multiple parties, a communication bottleneck arises for the party with labels, termed the *primary party*. This party must communicate with all other parties without labels, termed *secondary parties*, in each training round, placing substantial demands on the primary party's bandwidth. These challenges significantly hinder the practical deployment of VFL.

To address these issues, we introduce the *Federated Transformer (FeT)* to enhance performance and reduce privacy costs in multi-party fuzzy VFL. First, to tackle performance issues, we encode key similarities into data representations aligned by *positional encoding averaging*, which eliminates the need for quadratic calculations of key pairs. Additionally, we have designed a trainable *dynamic masking* module that automatically filters out incorrectly linked pairs, enhancing model accuracy by up to 13% in 50-party fuzzy VFL on the MNIST dataset. Second, to mitigate the escalating costs of privacy protection, we introduce SplitAvg, a hybrid approach that merges encryption-based and noise-based methods, maintaining a consistent noise level regardless of the number of participating parties. Third, to alleviate communication overhead on the primary party, we implement a *party dropout* strategy, which randomly excludes certain secondary parties during each training round. This effectively reduces communication costs by approximately 80% and improves model generalization. Our codes are available on GitHub[1]. In summary, our contributions are as follows:

- We design *Federated Transformer (FeT)*, a novel model achieving promising performance under multi-party fuzzy VFL.

- We introduce *SplitAvg* to enhance the privacy of FeT by protecting local representations in multi-party fuzzy VFL, with a theoretical proof of its differential privacy.

- Experimental results demonstrate that FeT significantly outperforms baseline models by up to 46% in terms of accuracy in 50-parties VFL. Moreover, while providing enhanced privacy, FeT consistently surpasses state-of-the-art models even in traditional two-party fuzzy VFL scenarios.

---

[1] https://github.com/Xtra-Computing/FeT

## 2 Preliminaries

In this section, we provide the foundational concepts necessary for understanding our approach to differential privacy. Differential Privacy (DP) offers a rigorous mathematical framework for preserving individual privacy. It quantifies privacy in terms of the probability of producing the same output from two similar datasets that differ by exactly one record.

**Definition 1.** *Consider a randomized function $\mathcal{M} : \mathbb{R}^d \to \mathcal{O}$ and two neighboring databases $D_0, D_1 \sim \mathbb{R}^d$ that differ by exactly one record. For every possible output set $O \subseteq \mathcal{O}$, $\mathcal{M}$ satisfies $(\varepsilon, \delta)$-differential privacy if*

$$\Pr[\mathcal{M}(D_0) \in O] \leq e^\varepsilon \Pr[\mathcal{M}(D_1) \in O] + \delta,$$

*where $\varepsilon \geq 0$ and $\delta \geq 0$.*

A single query that adheres to differential privacy is termed a *mechanism*. For example, the Gaussian mechanism [4] is commonly used to achieve DP by adding Gaussian noise to the output of the function.

**Theorem 1** (Gaussian Mechanism [4]). *For a function $f : \mathbb{X} \to \mathbb{R}^d$ characterized by a global $L_2$ sensitivity $\Delta_2$, which signifies that the maximum difference in the $L_2$-norm of the outputs of $f$ on any two neighboring databases is $\Delta_2$, and for any $\varepsilon \geq 0$ and $\delta \in [0, 1]$, the Analytic Gaussian Mechanism is defined as $\mathcal{M}(x) = f(x) + Z$, where $Z \sim \mathcal{N}(0, \sigma^2 \mathbf{I})$. This mechanism satisfies $(\varepsilon, \delta)$-differential privacy if $\Phi\left(\frac{\Delta_2}{2\sigma} - \frac{\varepsilon\sigma}{\Delta_2}\right) - e^\varepsilon \Phi\left(-\frac{\Delta_2}{2\sigma} - \frac{\varepsilon\sigma}{\Delta_2}\right) \leq \delta$, where $\Phi(t) = \frac{1}{\sqrt{2\pi}} \int_{-\infty}^{t} e^{-x^2/2} dx$ is the cumulative distribution function (CDF) of the standard univariate Gaussian distribution.*

When multiple queries are made on the same database, independent Gaussian noise is added to each query to maintain differential privacy. The privacy loss of the composition of Gaussian mechanisms is formulated in Theorem 2.

**Theorem 2** (Moments Accountant [1]). *There exist constants $c_1$ and $c_2$ so that given the sampling probability $q = \frac{L}{N}$ and the number of steps $T$, for any $\varepsilon < c_1 q^2 T$, DPSGD [1] is $(\varepsilon, \delta)$-differentially private for any $\delta > 0$ if we choose $\sigma > c_2 \frac{q\sqrt{T \log(1/\delta)}}{\varepsilon}$.*

## 3 Related Work

**Performance.** Traditional VFL approaches [29, 7] are typically limited to two-party scenarios. In contrast, existing multi-party VFL methods [12, 51, 33, 52, 38, 21, 26] often rely on the assumption of precise identifiers that ensure perfect alignment across all parties. These methods generally employ the SplitNN framework [45], where each party maintains a portion of the model, and the models are collaboratively trained on well-aligned data samples through the transfer of representations and gradients, commonly known as split learning. However, the requirement for perfect data alignment is impractical in many real-world scenarios [50, 3], where identifiers are often imprecise.

To address this limitation, semi-supervised VFL [22, 30, 19, 20] has emerged, attempting to improve model performance by leveraging unlinked records through semi-supervised or self-supervised learning. However, these methods still assume that datasets from each party can be precisely linked using exact identifiers, a premise that is often untenable in real-world settings [50, 3]. Given that the quality of linkage significantly impacts VFL accuracy [34], exploring effective linkage methods remains a pivotal issue in VFL.

On the other hand, FedSim [50], based on real linkage projects at the German Record Linkage Center (GRLC) [3], acknowledges that the keys of each party are usually not precisely linkable and that records may have one-to-many relationships, leading to fuzzy linkage scenarios, as seen with keys like GPS addresses. FedSim enhances training performance by performing soft linkage and conducting training based on transmitted key similarities. Nonetheless, it faces limitations in scalability beyond two parties and introduces new privacy concerns by directly transferring similarities.

In summary, while existing studies face significant performance challenges when handling fuzzy keys in multi-party settings, our proposed FeT demonstrates a scalable design that addresses these challenges and shows promising performance improvements in both multi-party fuzzy VFL and two-party settings compared to FedSim.

**Privacy.** The privacy concerns associated with VFL are multifaceted. First, the primary party may infer data representations from secondary parties [31]. Second, the secondary party may derive gradients from the primary party [41, 54]. Third, external attackers could conduct membership inference attacks [39] on the deployed model [51]. This paper primarily addresses the second concern: safeguarding representations, while acknowledging other concerns as open challenges.

To address the privacy of representations in VFL, various methods have been proposed, falling into two primary categories: encryption-based methods and noise-based methods. Encryption-based methods [26, 14, 33, 52, 38, 21, 36] utilize computationally intensive cryptographic techniques to encrypt intermediate results. However, these methods often incur significant communication overhead when scaled to multiple parties. Conversely, noise-based methods [47, 46] protect data by perturbing [47] or manipulating [46] local representations. These methods typically do not provide theoretical privacy guarantees or require substantial amounts of noise when scaling to multiple parties in VFL, which can degrade performance. Unlike existing studies that rely solely on either approach, this paper explores a combined strategy incorporating both encryption-based and noise-based methods, ensuring the model scales effectively to multiple parties without the need for excessive noise.

## 4  Problem Statement

In this section, we formalize the concept of multi-party fuzzy VFL. We consider a supervised learning task where one party holding labels, termed the *primary party $P$*, collaborates with $k$ parties that do not hold labels, referred to as the *secondary parties*. The primary party $P$ possesses $n$ data records denoted as $\mathbf{x}^P := \{x_i\}_{i=1}^n$ along with corresponding labels $\mathbf{y} := \{y_i\}_{i=1}^n$. Each secondary party $S_k$ maintains its own dataset $\mathbf{x}^{S_k}$. All parties share common features, referred to as identifiers, expressed as $\mathbf{x}^i = [\mathbf{k}^i, \mathbf{d}^i]$, where $[\cdot]$ signifies concatenation. These identifiers $\mathbf{k}^i$ may exhibit inaccuracies and fuzziness, despite residing within the same range.

$$\min_\theta \frac{1}{n} \sum_{i=1}^n \mathcal{L}(f(\theta; x_i^P, \mathbf{x}^{S_1}, \dots, \mathbf{x}^{S_k}); y_i) + \Omega(\theta)$$

In this formulation, $\mathcal{L}(\cdot)$ denotes the loss function, $\theta$ represents the model parameters, and $\Omega(\theta)$ refers to the regularization term. The symbol $n$ indicates the number of samples in the primary party $P$.

**Threat Model.** This study focuses on defending feature reconstruction attacks [25, 31], which target local representations shared with the primary party. FeT operates under the assumption that all participating parties are *honest-but-curious*, meaning they adhere to the protocol but may attempt to infer additional information about other parties. Furthermore, we assume that the parties do not collude with one another. While other forms of attacks, such as label inference attacks [13] and backdoor attacks aimed at compromising labels and gradients, exist, they are considered orthogonal to the objectives of this study. These additional threats will be explored in our future research.

## 5  Approach

In this section, we address the performance and communication challenges inherent in multi-party fuzzy VFL. To tackle these issues, we introduce a transformer-based architecture named the Federated Transformer (FeT). This model encodes keys into data representations, thereby reducing reliance on key similarities. To accurately exclude incorrectly linked data records, we propose a trainable dynamic masking module that generates masks based on keys. Furthermore, to combat overfitting caused by the large model and to alleviate communication bottlenecks faced by the primary party, we introduce a party dropout strategy that randomly invalidates some parties during training. Additionally, we identify a positional encoding misalignment issue across parties in the FeT and propose positional encoding averaging to ensure consistent alignment, thereby enhancing model performance.

### 5.1  Model Structure

The architecture of the FeT is illustrated in Figure 2. In FeT, each secondary party has an encoder, while the primary party has both an encoder and a subsequent decoder. The internal structure of both the encoder and decoder closely adheres to the conventional transformer model [43]. We utilize

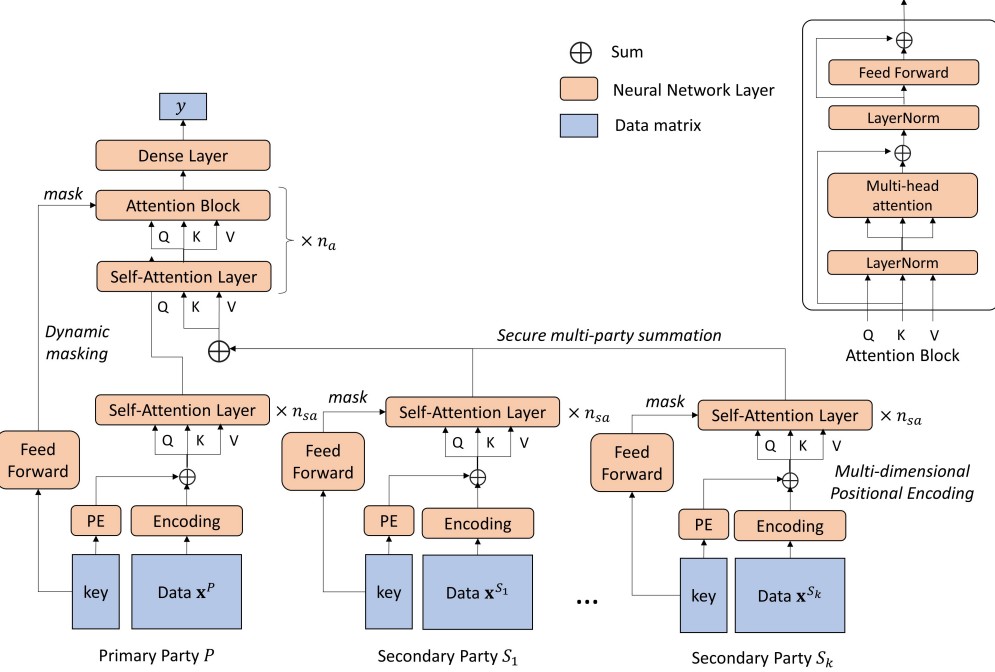

Figure 2: Structure of federated transformer (PE: multi-dimensional positional encoding)

multi-dimensional positional encoding [28] to integrate key information into feature vectors. Outputs from the encoders at the secondary parties are aggregated and then fed into the decoder. Details regarding the privacy mechanisms employed during this aggregation phase are discussed in Section 6, while the details of the training process are elaborated in Section 5.2. We then elaborate on three techniques designed to improve performance and reduce communication costs.

**Dynamic Masking.** The size of the neighborhood varies significantly depending on the party and key values. Consequently, including a large number of neighbors $K$ for all parties can hinder the model's ability to extract meaningful information and result in overfitting. To address this, we introduce a dynamic "key padding mask" in the transformer, generated from the identifier values using a trainable MLP. This approach allows the model to effectively disregard distant data records, thereby eliminating the influence of irrelevant data when $K$ is large. This concept resembles the weight gate in FedSim, but it diverges by using identifiers as inputs instead of similarities, enhancing privacy by preventing the transmission of similarity data across parties.

The learned dynamic masking is visualized in Figure 3. We derive two key insights from the visualization: (1) Dynamic masking effectively focuses on a localized area around the primary party's identifiers. Data records with distant identifiers on secondary parties (in cooler colors) receive small negative mask values, reducing their significance in the attention layers - without accessing the primary party's original identifiers. (2) The focus area varies in scale and direction across samples: for example, the left figure concentrates on a small bottom area, the middle figure on a small top area, and the right figure on a broad area in all directions.

**Party Dropout.** Extending the Federated Transformer (FeT) to support multiple parties can be challenging for several reasons. First, the communication bandwidth required by the primary party becomes a significant bottleneck within the SplitAvg framework, increasing linearly with the number of parties. Second, the inclusion of many parties can result in an excessive number of parameters, which may lead to overfitting. To mitigate these issues, we introduce the concept of *party dropout*. Inspired by traditional dropout [40], we randomly set a portion $r_d$ of the parties' representations to zero during training. This method serves as a form of regularization, thus helping to reduce overfitting, while also significantly cutting down on communication overhead. In our experiments, we demonstrate that increasing the party dropout rate to 0.8 leads to minimal accuracy loss and can

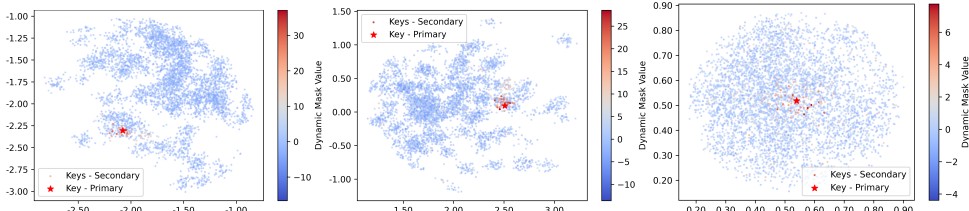

Figure 3: Learned dynamic masks of different samples: Each figure displays one sample (red star) from the primary party fuzzily linked with 4900 samples (circles) from 49 secondary parties. The position indicates the sample's identifier, and colors reflect learned dynamic mask values. Larger mask values signify higher importance in attention layers.

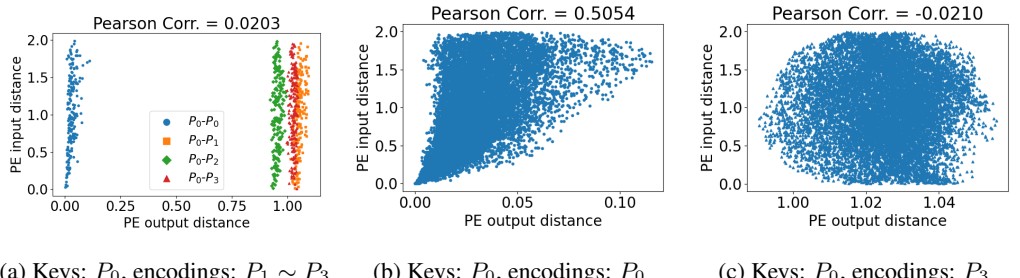

(a) Keys: $P_0$, encodings: $P_1 \sim P_3$     (b) Keys: $P_0$, encodings: $P_0$     (c) Keys: $P_0$, encodings: $P_3$

Figure 4: Misalignment of positional encoding ($P_0$: primary party; $P_1 \sim P_3$: secondary parties)

even improve accuracy. Consequently, the communication overhead on the primary party can be reduced by up to 80%, enhancing scalability when dealing with large numbers of parties.

Like traditional dropout, it is crucial to maintain consistent scaling of the representations during both training and testing phases. This consistency is naturally achieved within the SplitAvg framework. During the averaging process, if a ratio $r_d k$ of parties is set to zero, we adjust by dividing only by the number of non-zero parties, $(1 - r_d)k$. This method ensures that the scale of the averaged representations remains consistent across training and testing, regardless of the value of $r_d$.

**Positional Encoding Averaging.** In positional encoding (PE), it is generally expected that the distances between encoded representations are positively correlated with the distances between identifiers. In FeT, each party employs its own encoder and PE layer, each tasked with encoding its local identifiers into representations. This configuration leads to significant PE misalignment issues, as illustrated in Figure 4. Although the identifiers and their corresponding encoded representations maintain a positive correlation within the PE layer of each party, there is almost no correlation between the identifiers and encoded representations across different parties. This lack of correlation causes integration issues and affects accuracy. However, directly sharing a PE layer among all parties is not viable due to privacy concerns. To address this, we propose positional encoding averaging.

Every $T_{pe}$ epoch, the positional encoding layers are averaged and broadcast to all parties under a secure multi-party computation (MPC) scheme, akin to FedAvg [23] in horizontal federated learning [27]. While the privacy of the transmitted model can be a concern, this issue is an orthogonal open problem in horizontal federated learning.

## 5.2 Training

The FeT training process begins with employing Privacy-Preserving Record Linkage (PPRL) [44] to evaluate identifier similarities between the primary party $P$ and each secondary party. Secondary parties each contribute a random subset for linkage (line 5). For each $P$'s record, $K$ nearest neighbors within these subsets from secondary parties are determined (line 6). The training leverages data embeddings of dimensions $B \times L \times H$, where $B$ is batch size, $L$ is the sequence length, and $H$ is the hidden layer size, following the transformer architecture. In FeT's context, $L = 1$ for the primary and $L = K$ for secondary parties, linking each primary record with $K$ neighboring records from the secondaries. Identifiers are transformed into vectors using multi-dimensional positional

encoding [28] and combined with data representations for processing via self-attention blocks (lines 7, 10). Secondary parties' representations are averaged under the MPC protocol. The primary party then employs attention blocks for forward propagation to compute the final prediction (line 13). Backpropagation sends gradient updates from the primary to secondary parties to refine their local models (lines 14-16). The privacy mechanism including norm clipping (lines 8, 11) and distributed Gaussian noise (line 12) are further discussed in Section 6.

---

**Algorithm 1:** Training Process of Federated Transformer

**Input** :Primary party $\mathbf{x}^P$, secondary parties $\mathbf{x}^{S_1}, \ldots, \mathbf{x}^{S_k}$, label $\mathbf{y}$, noise scale $\sigma$, sampling ratio $q$, clipping threshold $C$

**Output** :Local models $\theta_l^P, \theta_l^{S_1}, \ldots, \theta_l^{S_k}$ and aggregation model $\theta_a^P$

1   Initialize model parameters $\theta_a^P, \theta_l^P, \theta_l^{S_1}, \ldots, \theta_l^{S_k}$

2   **for** *epoch $t \in [T]$* **do**

3     **for** *instance $x_i^P \in \mathbf{x}^P$ on primary party* **do**

4       **for** *party $h \in \{S_1, \ldots, S_k\}$* **do**

5         $\tilde{\mathbf{x}}^h \leftarrow$ randomly choose $q$ ratio from $\mathbf{x}^h$

6         $\tilde{\mathbf{x}}_i^h \leftarrow$ link $K$ records with $x_i^P$ from $\tilde{\mathbf{x}}^h$

7         $\tilde{\mathbf{R}}_i^h \leftarrow f(\theta_l^h; \tilde{\mathbf{x}}_i^h)$

8         $\hat{\mathbf{R}}_i^h \leftarrow \tilde{\mathbf{R}}_i^h / \max\left(1, \frac{\|k\tilde{\mathbf{R}}_i^h\|_2}{C}\right)$            // Norm clipping

9       **end**

10     $\mathbf{R}_i^P \leftarrow f(\theta_l^P; x_i^P)$

11     $\hat{\mathbf{R}}_i^P \leftarrow \mathbf{R}_i^P / \max\left(1, \frac{k\|\mathbf{R}_i^P\|_2}{C}\right)$            // Norm clipping

12     $\mathbf{H}_i \leftarrow \text{MPCAvg}\left(\mathbf{R}_i^{S_1}, \ldots, \mathbf{R}_i^{S_k}, \mathcal{N}(0, C^2\sigma^2)\right)$

13     $\hat{y}_i \leftarrow f(\theta_a^P; \mathbf{H}_i)$

14     $\nabla_{\theta_a^P} \leftarrow \frac{\partial \ell(y_i, \hat{y}_i)}{\partial \theta_a^P}$, $\theta_a^P \leftarrow \theta_a^P - \eta_t \nabla_{\theta_a^P}$

15     **for** *party $h \in \{P, S_1, \ldots, S_k\}$* **do**

16       $\nabla_{\theta_l^h} \leftarrow \frac{\partial \ell(y_i, \hat{y}_i)}{\partial \theta_a^h}$, $\theta_l^h \leftarrow \eta_t \nabla_{\theta_l^h}$

17     **end**

18    **end**

19   **end**

---

## 6   Privacy

In this section, we address the challenges of privacy in multi-party fuzzy VFL. First, the risk of transferring raw similarities has been mitigated by the design of the FeT itself. Second, to address the increasing costs when more parties join, we introduce a multi-party privacy-preserving VFL framework, *SplitAvg*, which is compatible with FeT. The architecture of SplitAvg is illustrated in Figure 5. SplitAvg integrates differential privacy (DP), secure multi-party computation (MPC) [6], and norm clipping to enhance the privacy of representations. Additionally, to further improve the utility of FeT under DP, we employ privacy amplification techniques that reduce the noise scales by incorporating random sampling.

### 6.1   Differentially Private Split Neural Network - SplitAvg

This section outlines three techniques applied to the SplitAvg to improve privacy: representation norm clipping, privacy amplification, and MPC with distributed Gaussian noise. These strategies collectively protect the privacy of each secondary party's representations through differential privacy and ensure that privacy risks do not escalate with an increase in the number of parties due to MPC.

**Representation Norm Clipping.** The magnitude of the $\ell_2$-norm is pivotal in determining the sensitivity of differential privacy. To limit the maximum change of the $\ell_2$-norm, norm clipping is essential. Specifically, for a representation $\mathbf{R}$, we ensure that $\|\mathbf{R}\|_2 \leq C$, where $C$ is a predefined positive

real number. This is achieved by scaling $\mathbf{R}$ by a factor of $C$, formally, $\hat{\mathbf{R}} = \mathbf{R}/\max(1, \|\mathbf{R}\|_2/C)$. Through this process, any representation $\mathbf{R}$ with $\|\mathbf{R}\|_2$ exceeding $C$ is scaled to $C$, while values of $\|\mathbf{R}\|_2$ below $C$ remain unaffected.

**Secure Multi-Party Computation with Distributed Gaussian Noise.** To address the challenges of applying differential privacy in multi-party VFL, we propose a method that integrates noise addition into the process of aggregating representations, facilitated through MPC. In this setup, each secondary party first independently conducts representation norm clipping to limit the scale of their data. Subsequently, these clipped representations, along with Gaussian noise $\mathcal{N}(0, \sigma/k^2)$ independently generated by each of the $k$ secondary parties, are aggregated through averaging under MPC. For the primary party, this aggregation is equivalent to adding independent Gaussian noise $\mathcal{N}(0, \sigma^2)$ to the averaged result. The adoption of MPC in our framework ensures that the secondary parties do not need to individually add $\mathcal{N}(0, \sigma^2)$ noise to their representations. Instead, as the primary party only has access to the averaged result under MPC, each secondary party can add a smaller amount of noise. This method effectively improves utility with a small efficiency cost due to MPC.

**Privacy Amplification by Secondary Subsampling.** This technique is specifically designed for FeT configurations. According to the principle of privacy amplification [5], applying a function to a randomly sampled subset of data enhances privacy compared to applying the same function to the entire dataset. By initiating linkage from a randomly sampled subset rather than the full dataset, the privacy parameters effectively shift from $(\varepsilon, \delta)$ to $(q\varepsilon, q\delta)$, where $q < 1$ represents the sampling rate. In FeT, the primary party typically selects subsets of candidate data records for training from each secondary party, targeting those with neighboring identifiers. By pre-sampling these subsets at a rate of $q$ before conducting a k-nearest neighbors (kNN) search, we avoid processing the entire dataset, which in turn reduces the noise required to maintain the same privacy level.

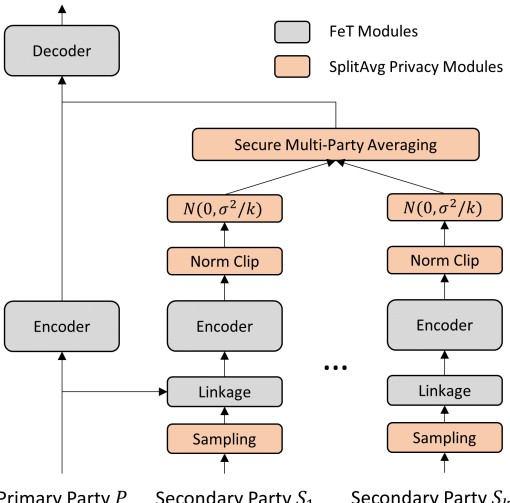

Figure 5: Differentially private split-sum neural network

## 6.2 Privacy Guarantee

Our analysis of differential privacy focuses on a hypothetical global dataset linked using all secondary parties, denoted as $\mathbf{x}^{S_1}, \ldots, \mathbf{x}^{S_k}$. Since these datasets are correlated, removing one data record from this global dataset will result in changes to the representations in all secondary parties. Consequently, privacy loss accumulates across secondary parties without the use of MPC. However, with MPC, a single aggregated noise, formed by distributing smaller noise contributions among parties, can be added, effectively reducing the overall required noise. The privacy guarantee for these representations is formally articulated in Theorem 3, with the proof provided in Appendix A.

**Theorem 3.** *For certain constants $c_1$ and $c_2$, given a sampling rate $q$, the total number of epochs $T$, and the number of batches $B$ in each epoch, each representation $\mathbf{R}^{S_k}$ achieves $(\varepsilon, \delta)$-differential privacy for all $\varepsilon < c_1 q^2 T$, with any $\delta > 0$, by selecting the standard deviation $\sigma$ of the Gaussian noise mechanism as follows:*

$$\sigma > c_2 \frac{q\sqrt{BT \log(1/\delta)}}{\varepsilon}. \tag{1}$$

## 7 Experiments

This section presents the experimental setup and results. We begin by outlining the experimental settings in Section 7.1, followed by an assessment of performance across varying numbers of parties and neighbors in Section 7.2. We then analyze the privacy of FeT in Section 7.3. Additionally, an ablation study is conducted in Appendix C to evaluate the contribution of each component to performance, including dynamic masking, party dropout, positional encoding, key fuzziness, and

SplitAvg. The performance of FeT with exact key matching is assessed in Appendix D, while the computational and memory efficiency of MPC and training is evaluated in Appendix E. Privacy evaluation on two-party real datasets is included in Appendix F. Furthermore, FeT's performance under imbalanced feature splits across parties, based on VertiBench [49], is presented in Appendix G.

## 7.1 Experimental Settings

**Datasets.** Our experiments utilize five datasets, including three real-world datasets: `house` [35, 2], `bike` [8, 42], and `hdb` [18, 37], along with two high-dimensional datasets: `gisette` [16] and `MNIST` [24]. Detailed descriptions of these datasets can be found in Appendix B. To simulate multi-party fuzzy VFL, we partition the features equally and randomly among the parties. The primary party's feature dimensions are reduced to 4 using principal component analysis (PCA) to serve as universal keys. To create fuzzy linked scenarios, we add independent Gaussian noise with a scale of 0.05 to the keys of each party.

**Baselines.** We include three baselines in our experiments: (1) Solo: training only on the primary party; (2) Top1Sim: linking each data record in the primary party only with its most similar neighbor in the secondary parties; (3) FedSim [50]: training on the top $K$ neighboring data records.

## 7.2 Performance

**Two-party fuzzy VFL.** In this experiment, we evaluate the performance of FeT in two-party settings without privacy mechanisms. The detailed results are presented in Table 1. Our experiments demonstrate that FeT consistently outperforms the leading two-party fuzzy VFL methods. Notably, this performance improvement is achieved while enhancing privacy protections, as FeT does not involve transferring similarity data.

Table 1: Root Mean Squared Error (RMSE) on real-world two-party fuzzy VFL datasets

| Algorithm | house | bike | hdb |
|-----------|-------|------|-----|
| Solo | 73.27 ± 0.16 | 244.33 ± 0.75 | 33.97 ± 0.61 |
| Top1Sim | 58.54 ± 0.35 | 256.19 ± 1.39 | 31.56 ± 0.21 |
| FedSim | 42.12 ± 0.23 | 235.67 ± 0.27 | 27.13 ± 0.06 |
| **FeT** | **39.75 ± 0.29** | **232.98 ± 0.62** | **26.94 ± 0.15** |

**Effect of Number of Neighbors $K$.** In this experiment, we assess the impact of the number of neighbors, $K$, on FeT's performance by varying $K$ from 1 to 100. The results are displayed in Figure 6. The FedSim baseline is trained using the optimal $K$ value (i.e., 50 for `hdb` and 100 for `house` and `bike`). The figure reveals two key insights: First, FeT's performance improves as $K$ increases, demonstrating its ability to filter useful information even as the number of unrelated data records grows. Second, FeT consistently outperforms all baselines at larger values of $K$, highlighting its superiority in fuzzy VFL scenarios.

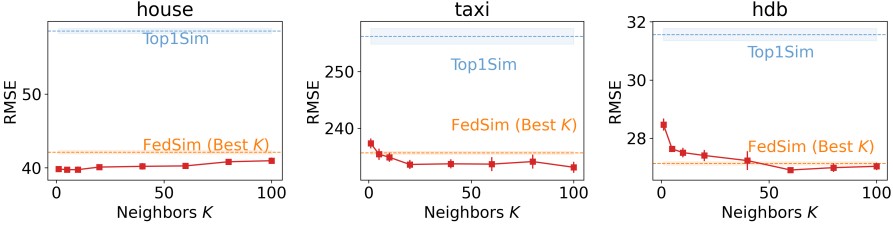

Figure 6: Effect of Different Number of Neighbors $K$ on FeT Performance

**Effect of Number of Parties** In this experiment, we assess FeT's performance in fuzzy VFL with various numbers of parties. Due to the absence of real multi-party VFL data, we employ synthetic data for our evaluations. We partition the features equally and randomly among the parties, reducing

the primary party's feature dimensions to 4 using PCA as the universal keys. To simulate fuzzy linked scenarios, we add independent Gaussian noise with a scale of 0.05 to the keys of each party.

Figure 7 shows that FeT generally outperforms the baselines, particularly with a larger number of parties. This advantage is attributed to Solo's lack of informative features and Top1Sim's noise-affected linkage. FedSim performs poorly as the top-1 linked secondary parties are unaware of the primary parties' keys, leading to misalignment in subsequent soft linkage and training steps. On the `gisette` dataset with $k = 10$, FeT and other models slightly underperform compared to Solo, likely due to overfitting given `gisette`'s small size.

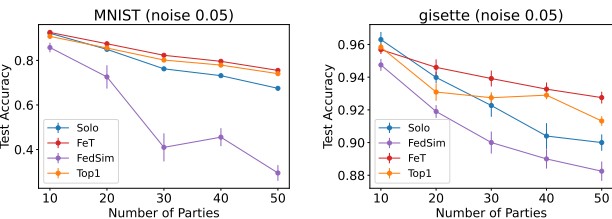

Figure 7: Impact of number of parties on FeT performance

## 7.3 Privacy

In this subsection, we analyze how the performance of FeT varies with different noise scales ($\sigma$) and sampling rates on secondary parties, demonstrating the impact of privacy constraints on its accuracy. The results are depicted in Figure 8. We observe three key points: First, a moderate sampling rate has a negligible effect on model performance and may even slightly improve performance (e.g., on the `MNIST` dataset) by reducing overfitting. Second, despite increasing noise levels and enhanced privacy guarantees, FeT consistently outperforms baseline models. Third, the $\varepsilon - \sigma$ privacy-noise curves illustrate that solely adding Gaussian noise without MPC, even with advanced analysis theorems such as Rényi Differential Privacy (RDP) [32], would require much larger noise scales compared to our approach that integrates MPC.

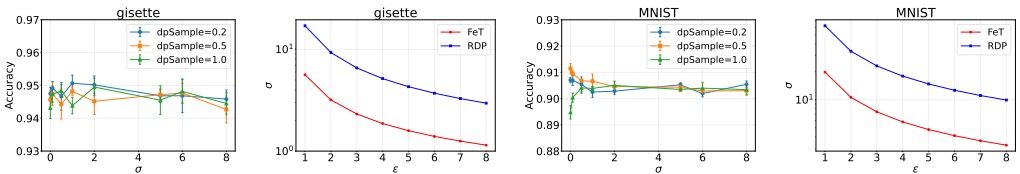

Figure 8: Impact of noise scale $\sigma$ on FeT accuracy and relationship between $\sigma$ and $\varepsilon$ under 10-party fuzzy VFL (RDP: without MPC, privacy loss calculated by Rényi differential privacy)

# 8 Conclusion

In this study, we introduce the Federated Transformer (FeT), specifically designed to support multi-party VFL while effectively addressing critical challenges related to performance, privacy, and communication. Furthermore, we provide theoretical proof of FeT's differential privacy, ensuring that data representations remain protected from secondary parties. Notably, our experiments demonstrate that FeT surpasses baseline models, even under stringent privacy guarantees and within the traditional two-party setting, establishing its efficacy and robustness in complex federated environments.

**Broader Impact.** The architecture of FeT, even without privacy mechanisms, has potential applications in multimodal learning. Multimodal tasks often require the alignment of data records across different modalities, which can be quite challenging. For instance, aligning 24Hz video streams with 48kHz audio tracks is complex, as each video frame may correspond to a range of audio samples. FeT has shown its capability to effectively learn from such fuzzily aligned data. Furthermore, the transformer model has proven effective across various data types, including images, text, and tabular data, highlighting FeT's suitability for multimodal learning applications.

## Acknowledgments

This research/project is supported by the National Research Foundation Singapore and DSO National Laboratories under the AI Singapore Programme (AISG Award No: AISG2-RP-2020-018), the National Research Foundation Singapore and Infocomm Media Development Authority under its Trust Tech Funding Initiative. Any opinions, findings and conclusions or recommendations expressed in this material are those of the author(s) and do not reflect the views of National Research Foundation Singapore, DSO National Laboratories, and Infocomm Media Development Authority. This research is also sponsored by Webank Scholars Program. We thank the AMD Heterogeneous Accelerated Compute Clusters (HACC) program (formerly known as the XACC program - Xilinx Adaptive Compute Cluster program) for the generous hardware donation.

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

# Appendix

## Table of Contents

## A   Proof

**Theorem 3.** *For certain constants $c_1$ and $c_2$, given a sampling rate $q$, the total number of epochs $T$, and the number of batches $B$ in each epoch, each representation $\mathbf{R}^{S_k}$ achieves $(\varepsilon, \delta)$-differential privacy for all $\varepsilon < c_1 q^2 T$, with any $\delta > 0$, by selecting the standard deviation $\sigma$ of the Gaussian noise mechanism as follows:*

$$\sigma > c_2 \frac{q\sqrt{BT \log(1/\delta)}}{\varepsilon}. \tag{1}$$

*Proof.* The proof leverages the moments accountant bound [1], which is applicable to a sequence of Gaussian mechanisms applied to subsampled data. We begin by establishing that the output's $\ell_2$-norm for each function is constrained by a constant $C$. This constraint ensures that each randomized function adheres to differential privacy under the Gaussian mechanism. By determining the cumulative count of Gaussian mechanisms applied, we can directly invoke Theorem 2 to reach our conclusion.

To clarify the process, we apply norm clipping to each party as specified in Section 6, scaling each party at a rate of $C/k$. This scaling guarantees that, for every $\hat{\mathbf{R}}_i^h$, the condition $\|\hat{\mathbf{R}}_i^h\|_2 \leq C/k$ is satisfied. Using the triangle inequality within normed vector spaces, we derive:

$$\|\mathbf{H}_i\|_2 = \left\|\sum_{h=1}^{k} \hat{\mathbf{R}}_i^h\right\|_2 \leq \sum_{h=1}^{k} \left\|\hat{\mathbf{R}}_i^h\right\|_2 = k \cdot \frac{C}{k} = C. \tag{2}$$

Since the $\ell_2$-norm of $\mathbf{H}_i$ is bounded by $C$, adding Gaussian noise to $\mathbf{H}_i$ satisfies the conditions for differential privacy. Throughout the training, $B \cdot T$ independent noises are introduced, resulting in a sequence of Gaussian mechanisms targeting a randomly selected subset at a ratio $q$. Consequently, Equation 1 is derived by directly applying Theorem 2. $\qquad\square$

# B   Experimental Details

**Datasets.**   In this section, we include the detailed information of each dataset used in the experiment. These real-world datasets are obtained in the same manner as those utilized in FedSim, with each dataset comprising two parties. Details of the real datasets are presented in Table 2. The synthetic dataset, gisette [16], consists of 6,000 instances and 5,000 features and serves as a binary classification task with balanced labels. The MNIST dataset [24] consists of 60,000 instances and 28x28 features for a 10-class digit classification task.

Table 2: Basic information of real-world VFL datasets

| Dataset | Primary Party (w/ labels) | | | Secondary Party | | | Identifiers | | Task |
|---|---|---|---|---|---|---|---|---|---|
| | #samples | #features | ref | #samples | #features | ref | #dims | type | |
| house | 141,050 | 55 | [35] | 27,827 | 25 | [2] | 2 | float | regression |
| bike | 100,000 | 6 | [8] | 200,000 | 964 | [42] | 4 | float | regression |
| hdb | 92,095 | 70 | [18] | 165 | 10 | [37] | 2 | float | regression |

**Metrics.**   For regression tasks, Root Mean Square Error (RMSE) is utilized, while accuracy is applied to classification tasks. Early stopping is performed based on the validation set, with the corresponding test scores reported.

**Hyperparameters.**   Each algorithm was run until convergence, with a maximum of 50 to 100 epochs. The learning rate and weight decay were consistently set at $10^{-3}$ and $10^{-5}$, respectively. For the Solo model, a multi-layer perceptron (MLP) with two hidden layers of 400 units each was employed. In contrast, the Top1Sim utilized a single-layer MLP with a hidden size of 200 for both local and aggregation models. For FedSim, the number of $K$ neighbors was selected from the set $\{50, 100\}$. For FeT, the number of blocks is set to 6 for both local model and aggregation model.

**Environments.**   We evaluate FeT on a server equipped with dual Intel Xeon Gold 6346 CPUs, eight A100 GPUs with CUDA version 12.2, and 1008GB RAM, running Python 3.10.13 with PyTorch 2.1.1+cu121 on Linux kernel 6.5.0. Efficiency experiments were conducted on a machine powered by a 64-core Intel(R) Xeon(R) Gold 6226R CPU @ 2.90GHz and 376 GB of RAM. Each experiment was run five times, and we report the average and standard deviation.

# C   Ablation Study

In this section, we evaluate the performance improvement of each proposed component of the FeT. Our findings indicate that **dynamic masking is crucial for enhancing performance, while both party dropout and positional encoding averaging contribute modestly to these improvements**. Detailed analyses are provided below.

## C.1   Dynamic Masking

We evaluate the performance of FeT with and without dynamic masking, as shown in Table 3. The evaluation includes two-party datasets (house, bike, hdb) and two 50-party synthetic datasets (MNIST and gisette). The results indicate that dynamic masking leads to an improvement of up to 13 percentage points, particularly noticeable in datasets with a large number of parties. This suggests that dynamic masking significantly enhances model performance, especially in multi-party settings.

## C.2   Party Dropout

Next, we evaluate the effect of the dropout rate under specific hyperparameter settings: the number of parties $k = 50$, the number of neighbors $K = 100$, and key noise set to 0.05. The impact of the party dropout rate is demonstrated in Figure 9 and Table 4. Our observations reveal that a moderate party dropout rate of 0.6 enhances FeT's generalization by reducing the model size. Notably, FeT maintains stable performance even as the dropout rate increases to 0.8. This indicates that party dropout not

Table 3: Effects of Dynamic Masking (DM) and Positional Encoding (PE) on FeT Performance

| Model | Datasets (metric) | | | | |
|---|---|---|---|---|---|
| | house (RMSE) | bike (RMSE) | hdb (RMSE) | MNIST (Accuracy) | gisette (Accuracy) |
| FeT w/o PE | 43.28 ± 0.74 | 234.25 ± 0.93 | 27.31 ± 0.23 | - | - |
| FeT w/o DM | 42.48 ± 0.45 | 236.26 ± 0.71 | 29.13 ± 0.18 | 72.89% ± 1.43% | 90.32% ± 0.52% |
| FeT | **39.75 ± 0.29** | **232.98 ± 0.62** | **26.94 ± 0.15** | **85.47% ± 0.13%** | **92.43% ± 0.24%** |

only improves generalization but also significantly reduces communication overhead across parties. Based on these findings, we set the dropout rate to 0.6 by default in multi-party experiments.

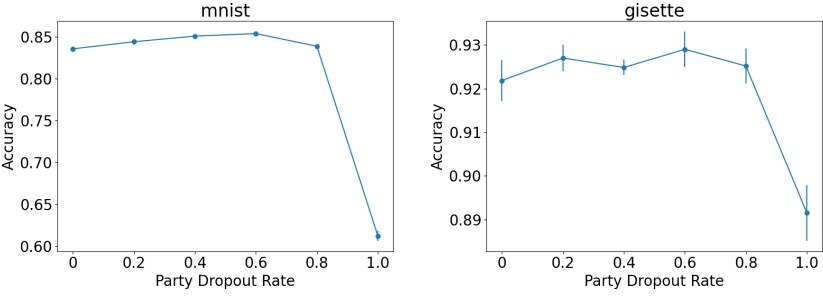

Figure 9: Effect of party dropout rate on FeT

Table 4: Effect of Party Dropout Rates on FeT Performance

| Dataset | Party Dropout Rate | | | | | |
|---|---|---|---|---|---|---|
| | 0 | 0.2 | 0.4 | 0.6 | 0.8 | 1.0 |
| gisette | 92.18% ± 0.47% | 92.70% ± 0.30% | 92.48% ± 0.18% | **92.90% ± 0.41%** | 92.52% ± 0.40% | 89.15% ± 0.64% |
| MNIST | 83.54% ± 0.30% | 84.39% ± 0.35% | 85.06% ± 0.17% | **85.36% ± 0.25%** | 83.85% ± 0.11% | 61.21% ± 0.58% |

### C.3 Positional Encoding

We now assess the effect of the frequency of positional encoding (PE) averaging, as depicted in Figure 10 and Table 5. We find that PE averaging yields improvements, particularly with a large number of parties, such as 50 on `MNIST`, where alignment of encodings becomes crucial. Based on our observations, we set the frequency to 1 in most experiments.

Additionally, we assess the impact of positional encoding on the performance of FeT, as detailed in Table 3. These evaluation of `MNIST` and `gisette` are conducted with the number of neighbors $K = 100$ and key noise 0.05. The results indicate that positional encoding is important for enhancing the performance of FeT.

### C.4 Fuzziness of Keys

We evaluate the impact of identifier fuzziness on FeT's performance by introducing Gaussian noise of varying scales to the keys. The results are presented in Figure 11. From the figure, we derive two key observations: (1) Both FeT and baseline models show improved performance in more balanced scenarios. (2) FeT consistently outperforms the baselines across different levels of heterogeneity, demonstrating its robustness to varying degrees of noise. These findings highlight the resilience of FeT in the presence of noise, which is critical for practical applications.

### C.5 SplitAvg vs. SplitNN

To evaluate the comparative performance of the SplitAvg (without noise) and SplitNN, we conducted training for both models using identical hyperparameters on the same VFL dataset, `gisette`. The

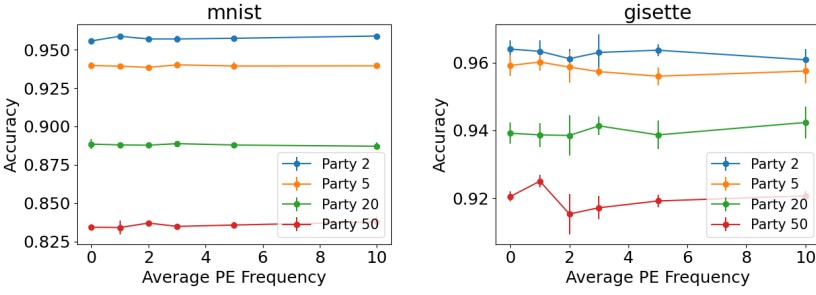

Figure 10: Effect of frequency of positional encoding averaging

Table 5: Ablation study for accuracy with different PE Average Frequency

| Dataset | #parties | PE Average Frequency | | | | | |
|---|---|---|---|---|---|---|---|
| | | 0 | 1 | 2 | 3 | 5 | 10 |
| gisette | 2 | **96.40% ± 0.25%** | 96.33% ± 0.33% | 96.12% ± 0.29% | 96.30% ± 0.53% | 96.37% ± 0.17% | 96.08% ± 0.32% |
| | 5 | 95.92% ± 0.31% | **96.02% ± 0.26%** | 95.87% ± 0.46% | 95.73% ± 0.12% | 95.60% ± 0.27% | 95.75% ± 0.37% |
| | 20 | 93.92% ± 0.31% | 93.87% ± 0.35% | 93.85% ± 0.60% | 94.13% ± 0.27% | 93.87% ± 0.42% | **94.23% ± 0.48%** |
| | 50 | 92.05% ± 0.15% | **92.50% ± 0.19%** | 91.53% ± 0.60% | 91.72% ± 0.34% | 91.92% ± 0.18% | 92.07% ± 0.14% |
| MNIST | 2 | 95.57% ± 0.12% | 95.88% ± 0.09% | 95.70% ± 0.16% | 95.70% ± 0.20% | 95.74% ± 0.13% | **95.89% ± 0.09%** |
| | 5 | 93.97% ± 0.22% | 93.92% ± 0.11% | 93.84% ± 0.18% | **94.01% ± 0.20%** | 93.94% ± 0.24% | 93.95% ± 0.09% |
| | 20 | 88.84% ± 0.31% | 88.79% ± 0.17% | 88.77% ± 0.13% | **88.88% ± 0.12%** | 88.79% ± 0.19% | 88.71% ± 0.24% |
| | 50 | 83.43% ± 0.16% | 83.41% ± 0.44% | 83.70% ± 0.09% | 83.48% ± 0.06% | 83.57% ± 0.22% | **83.78% ± 0.33%** |

outcomes are illustrated in Figure 12. Analysis of the figure yields two primary observations. Firstly, SplitNN and SplitAvg exhibit very similar loss and accuracy curves during training, indicating that both models behave very similarly. Secondly, upon expanding the number of participating parties to 128, we observe that the performance curves of both models remain closely aligned, albeit with the split-sum model exhibiting a marginally lower accuracy. This minor discrepancy is attributed to the increased model parameters in SplitNN, which can typically be compensated for by increasing the number of parameters in SplitAvg.

# D  Exact Linkage

While FeT primarily focuses on scenarios involving fuzzy linkage, we also evaluate its robustness in exact linkage contexts. To achieve this, we synthesize exact linkage data by generating pure random keys within the range of $[-1, 1]$ without introducing any noise. Each party is randomly divided into five or ten groups, with each group containing an equal number of features. Importantly, each party retains the exact keys, ensuring a controlled environment for our evaluation.

The results of our experiments are summarized in Table 6. From the table, we observe that Top1Sim achieves the highest accuracy, as it is inherently well-suited for exact linkage scenarios. In contrast,

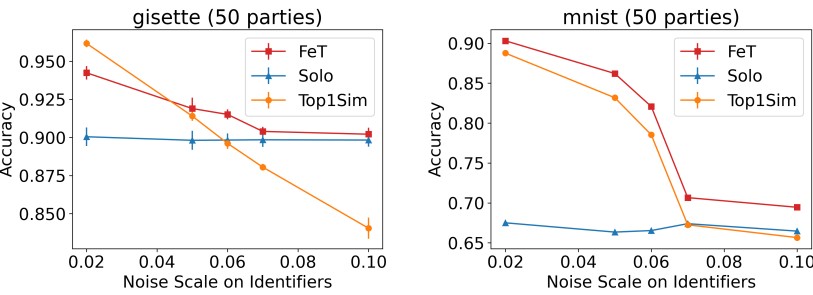

Figure 11: Effect of Fuzziness of Identifiers. The x-axis is the scale Gaussian noise added to precisely matched identifiers.

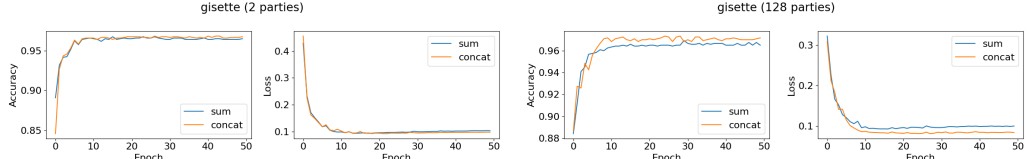

Figure 12: Test loss and accuracy curve of SplitAvg and SplitNN under same hyperparameters

the accuracy of FeT shows a slight decrease, which may be attributed to overfitting; however, its performance remains competitive and does not suffer significantly in this context.

Table 6: Performance of FeT under Exact Linkage

| Dataset | Algorithms | | |
|---|---|---|---|
| | Solo | Top1Sim | FeT (ours) |
| gisette (5-party) | 96.58% ± 0.25% | **97.52% ± 0.26%** | 94.73% ± 0.60% |
| MNIST (5-party) | 95.96% ± 0.03% | **96.96% ± 0.09%** | 96.53% ± 0.38% |
| gisette (10-party) | 96.30% ± 0.25% | **97.57% ± 0.25%** | 94.58% ± 0.41% |
| MNIST (10-party) | 92.09% ± 0.11% | 96.95% ± 0.06% | **96.97% ± 0.08%** |

# E   Efficiency

**Parameter Efficiency.**   In our analysis, we assess the computational efficiency of standard addition compared to multi-party computation (MPC) addition, as shown in Table 7. Under the arithmetic GMW protocol [15], and given that the size of the aggregated vector varies by dataset, we use a typical size for our experiments. Specifically, we conduct MPC addition to aggregate 10,000-dimensional vectors from multiple parties. Each experiment is performed five times, with the average timing reported. Although MPC generally incurs higher computational requirements, the results in Table 7 indicate that aggregating high-dimensional vectors via MPC incurs only a one-second overhead, even as the number of parties increases to 100. This minimal time cost is relatively small, especially when compared to other factors such as communication costs. Therefore, our findings suggest that MPC remains a feasible and efficient approach for representation aggregation in the context of VFL.

Table 7: Running time of summation with and without MPC in seconds

| #parties | Sum | MPC Sum | Overhead |
|---|---|---|---|
| 2 | $5.29 \times 10^{-6}$ | $5.09 \times 10^{-4}$ | $5.04 \times 10^{-4}$ |
| 5 | $2.10 \times 10^{-5}$ | $2.75 \times 10^{-3}$ | $2.73 \times 10^{-3}$ |
| 10 | $4.46 \times 10^{-5}$ | $1.02 \times 10^{-2}$ | $1.01 \times 10^{-2}$ |
| 20 | $1.29 \times 10^{-4}$ | $4.36 \times 10^{-2}$ | $4.36 \times 10^{-2}$ |
| 50 | $2.64 \times 10^{-4}$ | 0.268 | 0.268 |
| 100 | $5.21 \times 10^{-4}$ | 1.06 | 1.06 |

**Training Computational and Memory Efficiency.**   We evaluate the computational and memory efficiency of FeT during training on an RTX3090 GPU with a batch size of 128. The results, shown in Table 8, lead to three key observations: (1) FeT has a comparable number of parameters to FedSim; (2) FeT demonstrates improved memory efficiency compared to FedSim, although this improvement comes with a trade-off in training speed; and (3) the additional components, such as dynamic masking (DM) and positional encoding (PE), introduce only a minor overhead in terms of both parameters and computational cost.

Table 8: Training efficiency of FeT on RTX3090. PE: positional encoding; DM: dynamic masking.

| Dataset | #parameters | | | Train Seconds / epoch | | | Peak GPU Memory (MB) | | |
|---|---|---|---|---|---|---|---|---|---|
| | house | bike | hdb | house | bike | hdb | house | bike | hdb |
| FedSim | 3.47M | 1.85M | 1.87M | 9 | 38 | 6 | 2016 | 1917 | 1930 |
| FeT w/o PE | 0.98M | 3.24M | 0.63M | 35 | 50 | 15 | 397 | 691 | 539 |
| FeT w/o DM | 0.98M | 2.89M | 0.51M | 37 | 54 | 17 | 401 | 721 | 569 |
| **FeT** | 0.98M | 3.29M | 0.63M | 37 | 55 | 17 | 401 | 746 | 571 |

# F   Privacy on Two-Party Real Datasets

In this section, we explore how the performance of FeT varies with different noise scales $\sigma$ and secondary sampling rate, illustrating the influence of privacy constraints on its accuracy. The outcomes are depicted in Figure 13. From this figure, we observe two key points. First, for large secondary datasets like `bike`, a moderate sampling rate has a negligible effect on model performance. Conversely, for smaller secondary datasets like `hdb`, performance is quite sensitive to sampling rates. Second, as the noise scale increases for secondary parties, the performance of FeT does not degrade sharply; instead, it gradually converges to a state where only primary features are utilized due to our dynamic masking design. In this scenario, FeT also outperforms MLP-based Solo primarily due to the transformer's key encoding, which has proven to be more effective than incorporating all keys into the training process, as evidenced in spatial-temporal prediction tasks [9].

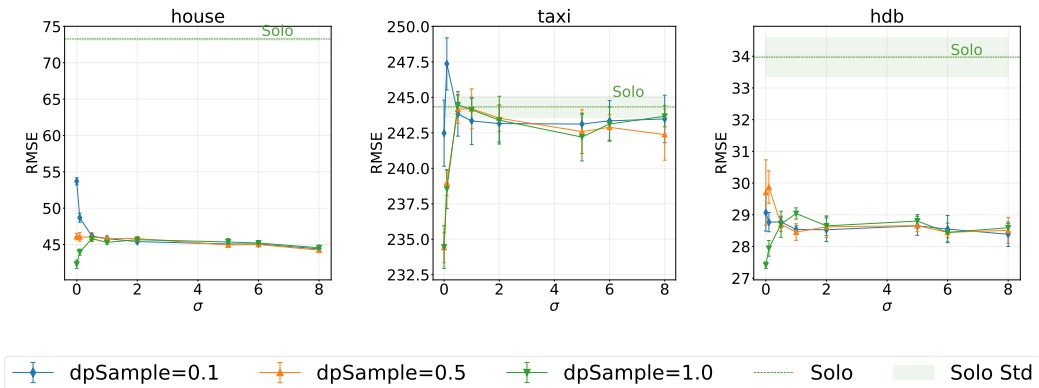

Figure 13: Impact of noise scale $\sigma$ on FeT performance

Next, we explore the relationship between $\sigma$ and $\varepsilon$ as outlined in Theorem 3, setting hyperparameters to reflect typical training conditions. The number of epochs is chosen based on common convergence epochs: 10 for `bike`, and 50 for `house` and `hdb`. We adopt a batch size of 8k and set $\delta$ to $1/N$, with $N$ representing the size of party $S_1$. This correlation between $\epsilon$ and $\sigma$ is depicted in Figure 14. The figure illustrates that reasonable noise levels can yield robust privacy guarantees. For instance, within a noise scale conducive to maintaining competitive performance, FeT achieves $\varepsilon = 3$ for `hdb` and $\varepsilon = 5$ for `house`, indicating effective privacy preservation under practical noise conditions.

# G   Performance on Imbalanced Split

The preceding experiments were conducted using a balanced feature split for VFL. Building on this foundation, we extended our evaluation of FeT to include datasets with varying levels of imbalance, motivated by the recent benchmarks presented in VertiBench [49]. The `MNIST` datasets are divided by features according to the methodology described in VertiBench [49], utilizing imbalance parameters $\alpha \in \{0.1, 0.5, 1.0, 5.0, 10.0, 50.0\}$, where a higher $\alpha$ value denotes greater balance across parties. The findings, illustrated in Figure 15, lead to two key observations: firstly, both FeT and the baseline algorithms exhibit improved performance in more balanced scenarios. Secondly, despite the varying

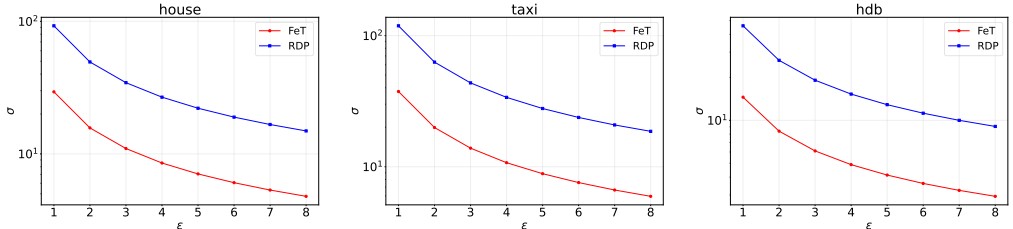

Figure 14: Relationship between $\varepsilon$ and noise $\sigma$

levels of data imbalance, FeT consistently shows competitive or superior performance relative to the baselines.

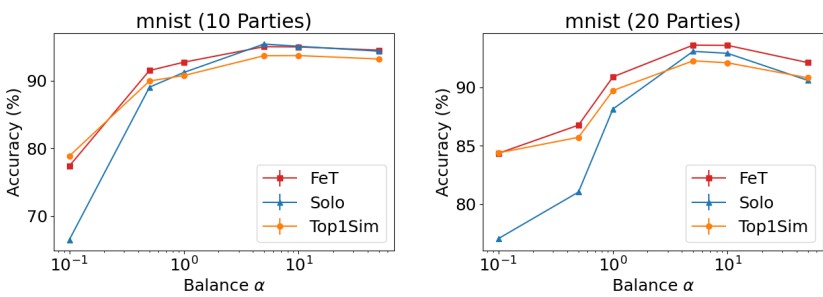

Figure 15: Performance on feature split with different level of imbalance

# H   Limitations

The design of FeT includes three primary limitations that warrant careful consideration. First, FeT operates under the assumption that common features exist across all parties. While this assumption is valid in many scenarios, it may not hold in more complex situations where the parties lack a shared set of features. This limitation necessitates further investigation into alternative frameworks or adaptations that can accommodate such cases, particularly in heterogeneous environments.

Second, although FeT facilitates the application of scalable differential privacy across multiple parties, the stringent privacy safeguards can lead to significant accuracy reductions when operating with low values of $\varepsilon$. This trade-off between privacy and utility is particularly concerning in performance-sensitive applications, where quantifying the extent of accuracy loss is essential for informing users about the potential impacts on their results. Future work should explore methods to balance privacy and accuracy more effectively.

Third, similar to other fuzzy VFL methods [50], FeT assumes a correlation between identifiers and data representations. This assumption may not hold in cases where identifiers are randomly generated, which could lead to overfitting and minor performance deficits compared to Top1 approaches. Experiments on such datasets (Appendix D) indicate that while FeT performs well in many scenarios, its effectiveness may vary significantly depending on the nature of the data and the key generation process. Therefore, further empirical studies are needed to assess FeT's robustness across diverse datasets and identifier generation strategies.

# I   License

The licenses of the datasets used in this work are presented in Table 9. We utilize the code from FedSim [50] as our baseline, which is licensed under the Apache V2 license[2]. Our own code will also be open-sourced under the Apache V2 license.

---

[2]`https://www.apache.org/licenses/LICENSE-2.0`

Table 9: Licenses of datasets

| Dataset | License | Adapt | Share | Commercial |
|---------|---------|:-----:|:-----:|:----------:|
| [35, 37] | CC0 1.0[a] | ✓ | ✓ | ✓ |
| [8] | NYCBS Data Use Policy[b] | ✓ | ✓ | ✓ |
| [2] | CC BY-NC-SA 4.0[c] | ✓ | ✓ | ✗ |
| [18] | Singapore Open Data License[d] | ✓ | ✓ | ✗ |
| [42] | All rights reserved | ✗ | ✗ | ✗ |

[a] https://creativecommons.org/publicdomain/zero/1.0/
[b] https://ride.citibikenyc.com/data-sharing-policy
[c] https://creativecommons.org/licenses/by/4.0/
[d] https://beta.data.gov.sg/open-data-license

