# OpenReview forum: "Federated Transformer: Multi-Party Vertical Federated Learning on Practical Fuzzily Linked Data"
_NeurIPS.cc/2024/Conference — NeurIPS 2024 poster_

### Official Review · Reviewer_P7Jg · 2024-07-11

**Soundness:** 2
**Presentation:** 2
**Contribution:** 2
**Rating:** 4
**Confidence:** 3

**Summary:**

The paper introduces the Federated Transformer (FeT), a novel framework for Vertical Federated Learning (VFL) that addresses the challenges of fuzzy identifiers in multi-party scenarios. It incorporates three key innovations to enhance performance, privacy, and reduce communication overhead: positional encoding averaging, dynamic masking, and a party dropout strategy. The approach significantly outperforms baseline models in scalability and privacy preservation.

**Strengths:**

1. The integration of a transformer architecture for managing fuzzy identifiers in multi-party VFL is innovative. The dynamic masking and party dropout strategies are creative solutions that enhance scalability and reduce communication costs.
2. The proposed model is well-articulated, with rigorous experimental validation that demonstrates substantial improvements over existing methods. The integration of differential privacy and secure multi-party computation strengthens the privacy aspect.
3.The paper is well-organized with clear explanations of the problems, proposed solutions, and results. It is accessible to readers with a background in federated learning.
4.The results show significant improvements in both performance and privacy, making it a valuable contribution to the field of federated learning, especially in applications involving sensitive data across multiple parties.

**Weaknesses:**

1.The performance improvements reported are impressive; however, how dependent are these improvements on the initial conditions of data alignment and the distribution of fuzzy identifiers? Could the authors provide insights on the robustness of the Federated Transformer (FeT) under less ideal conditions?
2. While the paper discusses scalability extensively, there is less focus on the computational resources required. Could the authors comment on the computational overhead and the practicality of deploying FeT in real-world scenarios with potentially limited resources?
3.The FeT introduces several complex mechanisms such as dynamic masking and positional encoding averaging. How do these additions affect the training time and complexity of the model? Is there a significant trade-off between performance and efficiency?
4.The experiments are conducted primarily on synthetic datasets. How well does FeT generalize to other real-world datasets, particularly those with higher levels of noise and less structured data?

**Questions:**

Please see Weakness and Limitations.

**Limitations:**

1.I think you can give us a better case diagram to help us better understand your new task.
2.I think there is an update error in line 16 of your algorithm.
3.In the Party Dropout section of the article, around line 188, you said that the communication overhead on the primary party can be reduced by up to 80%, but I did not find the corresponding experiment or proof in the article. I think this part needs to be better explained.
4.The experimental settings focus on synthetic and controlled environments. Real-world applications might introduce variables not accounted for in this study, potentially affecting the generalizability of the results.
5.The complexity of the model and the required computational resources are not thoroughly discussed, which could be a limitation for practical deployment in resource-constrained environments.
6.While the model addresses fuzzy identifiers, the performance heavily relies on the quality of the linkage, which may not always be feasible or accurate in real-world scenarios.

---

> ### Author Rebuttal · Authors · 2024-08-05
>
> Thank you for your valuable comments; we have addressed all concerns below.
>
> W1, L6. **How FeT Relies on Linkage Quality**: FeT is significantly **less reliant** on initial linkage quality than traditional VFL models like Top1Sim [4,5]. While Top1Sim trains only on records linked by privacy-preserving record linkage (PPRL) [3], FeT requires PPRL only to **approximate the K-nearest neighbors** and trains directly on these records, without needing them ordered by distance. Evidence for FeT's robustness includes:
> - **Ablation Study on K** (Figure 14, "rebuttal.pdf"): FeT consistently outperforms baselines with many unrelated data records included (K > 50), showing resilience to low linkage quality.
> - **Performance with Different Fuzziness** (Figure 12, "rebuttal.pdf"): FeT outperforms baselines with moderate to high noise in identifiers, demonstrating strong resilience to fuzziness.
> - **Dynamic Masking Visualization** (Figure 11, "rebuttal.pdf"): The visualization demonstrates that dynamic masking plays a crucial role in handling low-quality linkage, effectively focusing on a few close and relevant records among 4,900 fuzzily linked ones.
>
> **Performance of FeT in "Less Ideal" Cases**: FeT performs better in **general fuzzy VFL scenarios** but slightly underperforms Top1Sim [4,5] in **ideal cases with precisely matched identifiers**. This limitation is discussed in Appendix D with experiments on VFL datasets with randomly assigned precise IDs. Moreover, as shown in Figure 12 of "rebuttal.pdf," FeT only slightly underperforms compared to Top1Sim in low-noise conditions of `gisette` dataset.
>
> W2, W3, L5. **Why Focus on Performance Over Efficiency?**: In multi-party fuzzy VFL, the main challenge is fuzzy VFL models like FedSim [2] **perform worse than Solo training** in multi-party setting, instead of computational efficiency. FeT, as the first effective approach for multi-party fuzzy VFL, prioritizes performance to address this critical gap. For clarity, we will change the title to "_Federated Transformer: **Multi-Party** Vertical Federated Learning on Practical Fuzzily Linked Data_" and other related contents.
>
> **Efficiency of FeT**: We compared FeT's computational and memory efficiency with FedSim [2], the state-of-the-art fuzzy VFL model, as shown in Table 11 of "rebuttal.pdf," and identified three key findings:
>
> 1. **Parameter Efficiency**: FeT has a comparable or even smaller (23%-129%) number of parameters than FedSim, indicating that its performance improvement is due to model design rather than parameter scaling.
>
> 2. **Memory Efficiency**: FeT is significantly more memory efficient, consuming only 20-39% of the memory compared to FedSim. However, this efficiency comes at the cost of training speed. FeT performs neighbor search in parallel during real-time training, whereas FedSim spends hours linking top-K neighbors and preloads all linked neighbors into GPU memory, leading to longer linkage times, repeated data records, and higher memory usage.
>
> 3. **Overhead of New Components**: As detailed in Table 11 of "rebuttal.pdf," the additional components in FeT - dynamic masking and positional encoding - add minimal extra parameters (1k - 0.4M), causing only a slight computational overhead (0-5 seconds per epoch slowdown).
>
> In summary, FeT delivers **better performance and improved memory efficiency with a similar number of parameters** compared to FedSim, despite slightly lower training speed. Further optimization techniques, such as pipeline parallelism, could enhance FeT’s training speed but are beyond this study's scope.
>
> **VFL with Limited Resources**:
> 1. **FeT's Limited GPU Memory Performance**: Table 11 ("rebuttal.pdf") shows that FeT performs well even with GPU memory under 1 GB, making it viable for resource-constrained VFL scenarios.
>
> 2. **Cross-Silo VFL Prevalence**: Cross-device VFL with limited resources is rare. VFL typically occurs in cross-silo settings [3], involving dozens to hundreds of parties with adequate computational power. Discussions with industry collaborators providing commercial VFL services confirm that it's uncommon for a single user to have features spread across thousands of distinct parties.
>
> W4, L4. Three of our datasets (`house`, `taxi`, `hdb`) are real-world VFL datasets, identical to those used in FedSim [2]. Each party's data comes from different real sources; for example, the `taxi` dataset includes data from New York taxis and CitiBike. These datasets effectively demonstrate FeT's performance in real VFL applications.
>
> L1.  In Figure 15 of "rebuttal.pdf," we present a real-world application involving travel cost prediction in a city through collaboration among taxi, car, bike, and bus companies. Since personal travel information is private and cannot be shared, VFL is essential. Additionally, route identifiers - starting and ending GPS locations - can only be fuzzily linked, but linking closely related source and destination points with multi-party fuzzy VFL can significantly improve prediction accuracy.
>
> L2. We have fixed this typo in the revision.
>
> L3. The 80% communication reduction is straightforward from calculations, as the communication reduction is nearly proportional to the party dropout rate. Dropped parties don't participate in gradient and representation exchanges, which account for most communication in VFL.
>
> **References**
>
> [1] Li, et al. "Learnable fourier features for multi-dimensional spatial positional encoding." NeurIPS 21.
>
> [2] Wu et al. "A coupled design of exploiting record similarity for practical vertical federated learning." NeurIPS 22.
>
> [3] Vatsalan et al. "Privacy-preserving record linkage for big data: Current approaches and research challenges." Handbook of big data technologies, 17.
>
> [4] Hardy et al. Private federated learning on vertically partitioned data via entity resolution and additively homomorphic encryption. arXiv 17.
>
> [5] Nock et al. Entity resolution and federated learning get a federated resolution. arXiv, 18.

---

> > ### Comment · Reviewer_P7Jg · 2024-08-13
> >
> > Thanks for your response. According to your response and other reviewer's comments, I change the rating as "Borderline Reject".

---

### Official Review · Reviewer_so53 · 2024-07-12

**Soundness:** 3
**Presentation:** 2
**Contribution:** 3
**Rating:** 5
**Confidence:** 4

**Summary:**

This paper investigates vertical federated learning (VFL) linked with fuzzy identifiers and develops a new framework called Federated Transformer (FeT). The proposed framework leverages the transformer architecture to encode the identification information and distributes the subnets of the transformer across parties.

**Strengths:**

1.	The paper tackles the important topic of achieving multi-party fuzzy VFL with both promising performance and robust privacy.
2.	The design of using a transformer to encode fuzzy identifiers is quite novel.
3.	The code is publicly available.

**Weaknesses:**

1.	The paper is not well motivated. It would be great if the paper could provide concrete use cases of fuzzy VFL, that involves fuzzy data/linkages while requiring the privacy needs of distributed machine learning. The German Record Linkage Center might not be an appropriate example for federated learning.
2.	Several concepts in fuzzy VFL need further clarification. For example, what strategies are used in existing fuzzy linkage? How are the fuzzy identifiers presented? How is privacy preserved in fuzzy identifiers?
3.	The design rationale behind dynamic masking is unclear. Why is dynamic masking necessary? How does it function?
4.	The novelty of party dropout needs further elaboration. Dropout is widely used in federated learning. How does the proposed approach differ from existing designs?
5.	It is unclear why the proposed framework employs two different privacy mechanisms: differential privacy and MPC. What unique challenge is each privacy mechanism designed to address?

**Questions:**

1.	Please provide more concrete use cases of fuzzy VFL.
2.	Please clarify the key concept in fuzzy VFL.
3.	The design rationale behind the key components in the proposed framework is unclear. Please see my comments above and articulate them in more detail.

---

> ### Author Rebuttal · Authors · 2024-08-05
>
> Thank you for your valuable comments; we have addressed all concerns as follows.
>
> W1, L1. **Real Application of Multi-party Fuzzy VFL**: In Figure 15 of "rebuttal.pdf," we present a real-world application involving travel cost prediction in a city through collaboration among taxi, car, bike, and bus companies. Since personal travel information is private and cannot be shared, VFL is essential. Additionally, route identifiers - starting and ending GPS locations - can only be fuzzily linked, but linking closely related source and destination points with multi-party fuzzy VFL can greatly improve prediction accuracy.
>
> **German Record Linkage Center (GRLC) and VFL**: Previous VFL studies [1-3] show that record linkage is a necessary preprocessing step for VFL, making it inherently involved in all real VFL applications. While VFL studies are relatively new, record linkage has been extensively studied and widely applied. Thus, our study, along with [1], considers GRLC applications to be reflective of VFL applications.
>
> **Prevalence of Multi-party Fuzzy VFL in the Real World**: Traditional VFL assumes that each data record represents a **user**, making the assumption of a universal user ID natural. However, in real-world applications, these records often represent **various real-world objects** - such as a room (e.g., `house` and `hdb` datasets) or a travel route (e.g., `taxi` dataset). Expecting all real-world objects to have a universal, precise ID is unrealistic, which is why case studies in [1] found that over 70% of applications cannot be linked exactly.
>
> W2, L2. We adopt privacy-preserving record linkage (PPRL) [4] for similarity calculation. PPRL is a well-studied area separate from VFL, and we do not impose constraints on specific PPRL approaches, including linkage strategies, representation of fuzzy identifiers, or privacy mechanisms. For example, FEDERAL [5], a PPRL method, transforms fuzzy identifiers into Bloom filters that offer provable privacy guarantees. These Bloom filters' similarities reflect the original fuzzy identifiers' similarities. Our approach, like other fuzzy VFL algorithms [1-3], operates on these calculated similarities. While changing the PPRL method might impact the quality of the similarity measures, it does not affect FeT's superiority compared to other VFL algorithms.
>
> W3, L3. **Design Rationale**: Dynamic masking filters out unrelated data records with low similarities before they reach deep attention layers. FeT feeds the top-K similar records into the transformer, but this can introduce many unrelated records, leading to overfitting. To prevent this, a simple MLP creates a mask based on current identifiers, allowing the attention layers to focus on a narrower neighborhood and reducing overfitting.
>
> **Why Dynamic Masking is Necessary**: In our ablation study (Table 3 of Appendix C.1), removing dynamic masking (FeT w/o DM) results in significant performance loss across all five datasets. For example, on `MNIST`, training without dynamic masking led to a 13 percentage point drop in accuracy, demonstrating its necessity.
>
> **How Dynamic Masking Functions**: The dynamic masking module, a simple two-layer MLP, takes identifiers as input and outputs a mask added to the attention keys (used in `torch.nn.MultiHeadAttention` as `key_padding_mask`). The learned dynamic masks, visualized in Figure 11 of "rebuttal.pdf," reveal two key observations:
> - Dynamic masking effectively focuses on a localized area around the primary party's identifiers without accessing them directly. Records with distant identifiers on secondary parties (shown in cool colors) receive small negative mask values, reducing their significance in the attention layers.
> - The focus area varies in scale and direction across samples, indicating that the dynamic masking layer generates sample-specific masks to reduce overfitting.
>
> W4. Party dropout is a novel feature in VFL, distinct from traditional Dropout, introduced alongside our split-sum neural network design. **Almost all previous VFL** models [1-3,6] use a split-concat design, where representations are **concatenated** at the cut layer and passed to the aggregation model, requiring all parties to be present in each training step. In contrast, we use secure multi-party computation to **average** the representations, allowing some parties to be absent. This enables party dropout, which **disables training of entire encoders** of some parties in each step, rather than specific layers. Combined with SplitAvg, party dropout significantly reduces communication costs in multi-party VFL by the dropout ratio.
>
> W5. Our primary goal is to protect representations from secondary parties against an honest-but-curious primary party. We achieve this by applying differential privacy, adding noise to the representations. In VFL, unlike horizontal FL, data across parties are **related**. Thus, the primary party gains **access to more related representations as more parties join**. This typically requires adding more noise to maintain the same level of differential privacy, thus reducing utility. To address this, we use secure multi-party computation, letting the primary party know only the sum of the representations, keeping noise levels constant regardless of the number of parties.
>
> **References**
>
> [1] Wu et al. "A coupled design of exploiting record similarity for practical vertical federated learning." NeurIPS 22.
>
> [2] Nock et al. "The impact of record linkage on learning from feature partitioned data." ICML 21.
>
> [3]  Nock et al. Entity resolution and federated learning get a federated resolution. arXiv 18.
>
> [4] Vatsalan et al. "Privacy-preserving record linkage for big data: Current approaches and research challenges." Handbook of big data technologies, 17.
>
> [5] Karapiperis et al. "FEDERAL: A framework for distance-aware privacy-preserving record linkage." TKDE 18.
>
> [6] Liu, et al. "Vertical federated learning: Concepts, advances, and challenges." TKDE 24.

---

> > ### Comment · Reviewer_so53 · 2024-08-13
> >
> > Thank you for your responses. Most of my concerns have been addressed. I will raise my score.

---

### Official Review · Reviewer_sPbx · 2024-07-13

**Soundness:** 3
**Presentation:** 3
**Contribution:** 4
**Rating:** 7
**Confidence:** 4

**Summary:**

This paper introduces the Federated Transformer (FeT) framework, which is designed to support multi-party VFL.
It enhances the efficiency of model training among multiple parties using fuzzy identifiers while ensuring data privacy.
Experiment results show that FeT performed well when it scaled to 50 parties and the authors also provide theoretical proof of the differential privacy.

**Strengths:**

**Strengths of the Paper**

**Originality**

The biggest novelty of the work is the introduction of the Federated Transformer (FeT) framework. It presents a new aspect of handling vertical federated learning (VFL) with fuzzy identifiers.

- **FeT Structure**: The work introduces the architecture of FeT in detail. The model can be split into two parties: the primary party and the secondary party. It also includes several innovative techniques including Dynamic Masking, Party Dropout, and Positional Encoding Averaging.

**Experiment**

The authors compare the performance of the Federated Transformer (FeT) against multiple baseline methods which is a robust benchmarking. The experimental results demonstrate that FeT significantly outperforms baseline models, achieving improvements of up to 46 percentage points when scaled to 50 parties, which is a significant performance gain.

**Significance and Future Impact**
FeT is suitable for multimodal learning which aligns many federated learning scenarios in real life while protecting individual privacy.

**Weaknesses:**

**Weaknesses of the Paper**

**Model Design**

1) The introduction of a trainable dynamic masking module aims to improve the exclusion of incorrectly linked data records. However, if this module is not well-optimized, it could introduce additional noise or errors which degrades the model performance.

2) The proposal of positional encoding averaging may not available in all VFL scenarios, which may cause inconsistencies in data processing which may affect the model training.

**Privacy Trade-offs**

The paper says that stringent privacy safeguards may cause accuracy reductions, particularly with low values of $\epsilon$. It would be better if the authors could provide a more detailed analysis of the trade-offs between privacy and performance.

**Theoretical Proof**

While the paper provides some theoretical proofs, it may not offer a complete set of proofs for all claims made. The theoretical section did not adequately discuss the limitations of the proposed methods or the implications of violating the underlying assumptions. A more thorough exploration of these aspects would enhance the robustness of the theoretical framework.

**Experiment**

The experiments are conducted on a small number of datasets, which may not represent the full range of scenarios in VFL. Furthermore, The experiments explore little the trade-off between privacy guarantees and model utility, especially at low values of ε in differential privacy.

**Questions:**

**Questions**

1\. **Scenario Application**:

How can FeT be adapted to work with unstructured data types such as text or images? Is there any preliminary results or insights on applying FeT to these types of data?

2\. **Privacy-Performance Trade-off**:
Can you provide a more detailed analysis of the trade-off between privacy and performance, especially at different levels of $\epsilon$?

3\. **Model Performance on datasets with different features**:
How does the performance of the Federated Transformer (FeT) vary when applied to datasets with significantly different feature distributions across multiple parties?

**Limitations:**

The authors have addressed the limitations of their work by identifying three primary limitations related to the assumptions of common features across parties, the potential accuracy reductions due to stringent privacy safeguards, and the correlation between keys and data representations.

Here are some other potential limitations that may be considered:

**Large scale cases**: The model's performance may have the risk of degrading with a large number of parties or larger datasets, which may limit its application in certain large federated learning scenarios.

**Privacy Trade-offs**: While the model is designed to enhance privacy, there may still be some vulnerabilities to be exploited, especially when the underlying assumptions about data sharing are violated.

---

> ### Author Rebuttal · Authors · 2024-08-05
>
> Thank you for your valuable comments; we have addressed all concerns as follows.
>
> W1.1 **Design of Dynamic Masking**: Dynamic masking is a simple two-layer MLP that can be easily optimized and performs well across all five datasets. Our ablation study in Table 3 of Appendix C.1 demonstrates that dynamic masking **consistently and significantly improves performance on all five datasets**. For example, on the `MNIST` dataset, omitting dynamic masking results in a 13 percentage point drop in accuracy. To provide further insight, we **visualize the learned dynamic mask in Figure 11** of the attached "rebuttal.pdf."
>
> This visualization reveals two key observations: First, dynamic masking effectively focuses on a localized area around the identifiers of the primary party. Data records with more distant identifiers on secondary parties (shown in cooler colors) are assigned small negative mask values, reducing their influence in the attention layers. Second, the focus area varies in scale and direction across different samples, indicating that the dynamic masking layer learns to generate sample-specific masks, which helps to reduce overfitting.
>
> W1.2 **Design of Positional Encoding (PE) Averaging**: This design is specifically applicable to VFL scenarios with independent and identically distributed (i.i.d.) identifiers, similar to the FedAvg approach. VFL with heterogeneous or non-i.i.d. identifiers (e.g., one party using GPS coordinates while another uses postal codes) remains an open problem, posing significant challenges for identifier alignment - an area not yet explored in VFL research. Our ablation study in Table 5 of Appendix C.3 shows that PE averaging improves performance compared to non-PE-averaging FeT (average frequency = 0) in most datasets.
>
> W2 **Privacy Trade-offs**: The trade-off between privacy and performance is illustrated in Figure 5 and Figure 9 of the paper. Both performance (measured by accuracy) and privacy (denoted by $\varepsilon$) are influenced by the noise scale $\sigma$. Generally speaking, $\varepsilon$ reflects the probability bound of determining whether a specific data record exists in the training set based on the representations, regardless of the attack method. Thus, a lower $\varepsilon$ indicates higher privacy. To achieve high privacy, such as $\varepsilon = 1$, a noise scale of approximately $\sigma \approx 8$ is required, which significantly reduces utility, causing FeT to perform similarly to Solo. Conversely, when $\varepsilon$ is larger, such as $\varepsilon = 8$, the required noise scale decreases to $\sigma = 1$, which allows FeT to maintain relatively good performance.
>
> W3. **Theoretical Proof**: The full proof of Theorem 3 is provided in Appendix A. Background theorems and definitions of differential privacy are covered in Section 2, while the threat model is detailed in Section 4. Since Theorem 3 is based on differential privacy, its assumptions inherit the assumptions of differential privacy.
>
> W4 **Experiments on More Scenarios of VFL**: We have added experiments using the VertiBench [1] datasets to cover a broader range of VFL scenarios, as detailed in the attached "rebuttal.pdf." The results demonstrate that: 1) FeT outperforms baselines across varying levels of imbalance between parties, and 2) FeT shows better performance in VFL scenarios where parties have more balanced features.
>
> Q1. **Scenario Application**: Images and texts can be effectively handled by transformer structures, as demonstrated in many existing works [2]. Thus, adapting FeT to other unstructured data should not pose a significant challenge. The two main ideas - 1) encoding common features as positions and 2) dynamic masking -are particularly useful for multimodal alignment in these scenarios. The primary challenge, however, lies in the significant heterogeneity of features across parties. As shown in Figure 13 of the "rebuttal.pdf," although FeT outperforms baselines, its performance on imbalanced features requires further improvement. We are actively working on addressing this issue.
>
> Q2. Please refer to W2. Privacy Tradeoffs.
>
> Q3. **Significant Heterogeneous Features** We have conducted additional experiments using the VertiBench [1] datasets with varying heterogeneity (balance level $\alpha$) across parties, as shown in Figure 13 in the attached "rebuttal.pdf." The results demonstrate that while FeT's absolute performance decreases under severely heterogeneous feature distributions (very low $\alpha$), its improvement over Solo training becomes more pronounced in such heterogeneous scenarios.
>
>
> L1. **Large Scale Cases**: Unlike horizontal federated learning, Vertical Federated Learning (VFL) is primarily observed in cross-silo scenarios [3] rather than cross-device scenarios [4], as it is unlikely that a single user would have features distributed across thousands or even millions of distinct parties. This observation is supported by our communication with industry collaborators that provide commercial VFL service. Therefore, we focus on the cross-silo case, where collaborations among dozens or hundreds of parties are more common.
>
> L2. **Privacy Trade-offs**: We agree that malicious parties, rather than our assumed honest-but-curious parties, would present a greater privacy risk to the model design. We will acknowledge this limitation and plan to explore such scenarios in future work.
>
> **References**
>
> [1] Wu et al. "VertiBench: Advancing feature distribution diversity in vertical federated learning benchmarks." ICLR 24.
>
> [2] Han et al. "A survey on vision transformer." TPAMI 22.
>
> [3] Huang et al. "Cross-silo federated learning: Challenges and opportunities." arXiv 22.
>
> [4] Karimireddy et al. "Breaking the centralized barrier for cross-device federated learning." NeurIPS 21.

---

### Official Review · Reviewer_jPVu · 2024-07-16

**Soundness:** 3
**Presentation:** 3
**Contribution:** 2
**Rating:** 6
**Confidence:** 4

**Summary:**

This paper proposes the Federated Transformer (FeT) framework, which aims to address performance and privacy challenges in multi-party fuzzy vertical federated learning (VFL). FeT leverages the Transformer architecture to encode fuzzy identifiers and distribute training across different parties. The authors introduce three innovative techniques—position encoding averaging, dynamic mask module, and party dropout strategy—to enhance model performance while minimizing computational and communication overhead. Additionally, FeT incorporates a scalable privacy framework that combines differential privacy and secure multi-party computation, effectively safeguarding local data representations and ensuring manageable privacy maintenance costs. Experimental results demonstrate that FeT outperforms baseline models, offering superior performance and enhanced privacy protection. Overall, FeT overcomes the limitations of existing models in multi-party fuzzy VFL, showcasing exceptional performance and practicality.

**Strengths:**

1. This manuscript proposes the FeT framework. The framework combines federated learning with the Transformer architecture. At the same time, the dynamic mask technique, participant discarding mechanism, and location coding average techniques are applied to improve the model's performance and privacy protection in multi-party fuzzy vertical federation learning.

2. The FeT framework introduces SplitAvg, a hybrid privacy protection mechanism, which effectively reduces the introduction of noise and improves data utility. In addition, the privacy amplification technology enhances the privacy protection effect. This ensures data privacy as well as the efficiency and robustness of the model.

**Weaknesses:**

This manuscript proposes a scalable vertical federated learning for practical fuzzy linked data, aiming to solve the performance and privacy issues of real-world VFL in its application. However, there are still some major issues, as follows:

1. In the Introduction section, the definition of fuzzy identifiers and multi-party fuzzy VFL and the relationship between the identifiers, fuzzy identifiers and multi-party fuzzy VFL is not clearly introduced, which may lead to confusing concepts for readers. In addition, the important role of fuzzy identifiers in linking real-world data sets should be emphasized to prove the practical value of studying multi-party fuzzy VFL.

2. In the training of FeT, PPRL was first used to evaluate the identifier similarity between the primary participant P and each secondary participant. However, the author did not mention this method in the previous content. Based on this, a very important question is: in the comparative experiment, the author did not seem to compare the performance gap, privacy gap, and computing resource consumption gap between the training using only the PPRL method and the training based on the transformer structure in this article?

3. In FeT, the combination of dynamic mask, participant discarding mechanism, and hybrid privacy protection mechanism will inevitably affect the training process's time, computing resource consumption, and privacy protection ability. Therefore, it is also important to compare the training efficiency and computing resource consumption, as well as the privacy protection ability of FeT methods with current mainstream methods. This represents whether FeT can surpass other methods in real-world applications. The setting of the experiment may be as follows: 1. Training time of FeT and mainstream methods on the same data set and under the same conditions. 2. CPU/GPU usage between FeT and mainstream methods in the training process to measure memory consumption and network communication overhead, especially in the case of multi-party participation. 3. Evaluate the privacy protection effect of FeT and mainstream methods through quantified privacy protection indicators (such as ε value of differential privacy).

Minor weaknesses:
1. In lines 27 and 28, the authors describe collaboration between hospitals, financial institutions, and sensors. It is equivalent to juxtaposing the three, which is unreasonable.
2. In lines 58 and 59, the authors mention that experiments have shown that model accuracy has been improved by up to 13 percentage points on a 50-square fuzzy VFL on the MNIST dataset. It is not given here which method is compared with, please add.
3. In line 139, the author defines a common feature shared by all parties as an identifier, expressed as \(x^i = [k^i, d^i] \), where \([\cdot]\) signifies. \([,] \) and\([\cdot]\) is not the same

**Questions:**

same as the weakness

---

> ### Author Rebuttal · Authors · 2024-08-05
>
> Thank you for your valuable comments; we have addressed all concerns as follows.
>
> W1. We will include a formal definition in Section 4 and the introduction. Our scenario extends the two-party fuzzy VFL model defined in FedSim [1] to a multi-party setting. Below, we clarify the key terms:
>
> - **Multi-party VFL**: Federated learning tasks involving $k+1$ parties, denoted as $\{S_h\}_{h=0}^k$. Each party $S_h$ has $n_h$ data records, represented as $\mathbf{x}^{S_h} \in \mathcal{R}^{n_h \times (c + m_h)}$, where $c$ is the number of common features shared by all parties, and $m_h$ refers to the unique features held by each party $S_h$.
>
> - **Identifier**: For each data instance $x_i^{S_h} \in \mathbf{x}^{S_h}$, the $c$ common features $k_i^{S_h} \in \mathcal{R}^{1 \times c}$ are termed the **identifier** or **key** of $x_i^{S_h}$. All identifiers for $S_h$ are denoted as $\mathbf{k}_i^{S_h} := \{k_i^{S_h}\}_{i=1}^{n_h}$.
>
> - **Fuzzy Identifier**: In multi-party VFL, if two parties $S_a$ and $S_b$ have identifiers such that $\forall k_i \in \mathbf{k}_i^{S_a}$ and $\forall k_j \in \mathbf{k}_j^{S_b}$, $k_i \neq k_j$, then $\mathbf{k}_i^{S_a}$ and $\mathbf{k}_j^{S_b}$ are **fuzzy identifiers**.
>
> - **Multi-party Fuzzy VFL**: A VFL scenario involving parties with fuzzy identifiers.
>
> **Importance of Fuzzy Identifiers**: The critical role of linkage quality in VFL performance has been shown both empirically [1] and theoretically [2]. This impact is even more pronounced in multi-party settings with increasing noise, as demonstrated by our significant improvement over FedSim in such scenarios.
>
> **Prevalence of Multi-party Fuzzy VFL in the Real World**: Traditional VFL assumes that each data record represents a **user**, making the assumption of a universal user ID natural. However, in real-world applications, these records often represent **various real-world objects** - such as a room (e.g., `house` and `hdb` datasets) or a travel route (e.g., `taxi` dataset). Expecting all real-world objects to have a universal, precise ID is unrealistic, which is why case studies in [1] found that over 70% of applications cannot be linked exactly.
>
>
> W2. Privacy-Preserving Record Linkage (PPRL) is a linkage algorithm that calculates identifier similarities but lacks learning mechanisms, serving as a preprocessing step for most VFL algorithms. As such, PPRL is not directly comparable to VFL approaches like FeT.
>
> An extension of PPRL, known as Top1Sim [3,4], performs VFL on the most similar records identified by PPRL. We have thoroughly compared Top1Sim as a baseline. However, this simple extension overlooks significant information, which led to the development of learning-integrated approaches like FedSim [1] and FeT. Notably, Top1Sim incurs a 42% higher RMSE compared to FeT on the `house` dataset.
>
>
> W3. **Efficiency of FeT**: We compared FeT's computational and memory efficiency with FedSim, the state-of-the-art fuzzy VFL model, as shown in Table 11 of "rebuttal.pdf," and identified three key findings:
>
> 1. **Parameter Efficiency**: FeT has a comparable or even smaller (23%-129%) number of parameters than FedSim, indicating that its performance improvement is due to model design rather than parameter scaling.
>
> 2. **Memory Efficiency**: FeT is significantly more memory efficient, consuming only 20-39% of the memory compared to FedSim. However, this efficiency comes at the cost of training speed. FeT performs neighbor search in parallel during real-time training, whereas FedSim spends hours linking top-K neighbors and preloads all linked neighbors into GPU memory, leading to longer linkage times, repeated data records, and higher memory usage.
>
> 3. **Overhead of New Components**: As detailed in Table 11 of "rebuttal.pdf," the additional components in FeT - dynamic masking and positional encoding - add minimal extra parameters (1k - 0.4M), causing only a slight computational overhead (0-5 seconds per epoch slowdown).
>
> In summary, FeT delivers **better performance and improved memory efficiency with a similar number of parameters** compared to FedSim, despite slightly lower training speed. Further optimization techniques, such as pipeline parallelism, could enhance FeT’s training speed but are beyond this study's scope.
>
> **Privacy of FeT**: Figures 5(b, d) and 10 in the paper show FeT's privacy superiority. Since related studies use similar Gaussian noise mechanisms, we compare the noise scale at the same $\varepsilon$ rather than accuracy. At the same $\varepsilon$, FeT requires much less noise than RDP, a previous pure DP approach like FedOnce [5], generally leading to better model utility.
>
> MW1. **Real Application of Multi-party Fuzzy VFL**: In Figure 15 of "rebuttal.pdf," we present a real-world application involving travel cost prediction in a city through collaboration among taxi, car, bike, and bus companies. Since personal travel information is private and cannot be shared, VFL is essential. Additionally, route identifiers - starting and ending GPS locations - can only be fuzzily linked, but linking closely related source and destination points with multi-party fuzzy VFL can significantly improve prediction accuracy.
>
> MW2. The 13% improvement comes from comparing FeT to FeT without dynamic masking, as shown in Table 2 of Appendix C.1.
>
> MW3. We will correct this formulation in the revision.
>
> **References**
>
> [1] Wu et al. "A coupled design of exploiting record similarity for practical vertical federated learning." NeurIPS 22.
>
> [2] Nock et al. "The impact of record linkage on learning from feature partitioned data." ICML 21.
>
> [3] Hardy et al. Private federated learning on vertically partitioned data via entity resolution and additively homomorphic encryption. arXiv, 17.
>
> [4]  Nock et al. Entity resolution and federated learning get a federated resolution. arXiv, 18.
>
> [5] Wu et al. "Practical vertical federated learning with unsupervised representation learning." IEEE Transactions on Big Data, 2022.

---

> > ### Comment · Reviewer_jPVu · 2024-08-14
> >
> > Thanks for your responses to address my concerns.

---

### Author Rebuttal · Authors · 2024-08-05

We sincerely thank all the reviewers for their thoughtful efforts in reviewing our manuscript and providing valuable feedback. In response, we have made substantial revisions and added new experiments, visualizations, and real-world applications to the "rebuttal.pdf," including Figures 11-15 and Table 11. Below is a summary of the contents and key findings for each figure and table:

- **Figure 11**: Visualization of learned dynamic masks for different samples. Each figure displays one sample (red star) from the primary party, fuzzily linked with 4,900 samples (circles) from 49 secondary parties. The position indicates the sample's identifier, and colors reflect the learned dynamic mask values, with larger mask values indicating higher importance in the attention layers.
  - **Findings**:
    1. Dynamic masking effectively focuses on a localized area around the primary party's identifiers. Data records with distant identifiers on secondary parties (shown in cooler colors) receive small negative mask values, reducing their significance in the attention layers - without accessing the primary party's original identifiers.
    2. The focus area varies in scale and direction across samples: for example, the left figure concentrates on a small bottom area, the middle figure on a small top area, and the right figure on a broad area in all directions.

- **Figure 12**: Analysis of the effect of identifier fuzziness, showing the scale of Gaussian noise added to precisely matched identifiers.
  - **Findings**: FeT consistently outperforms baselines at moderate fuzzy scales. When identifiers are very noisy, FeT's performance approaches that of Solo. Conversely, when identifiers are highly accurate, FeT's performance approaches that of Top1Sim.

- **Figure 13**: Performance on VertiBench MNIST datasets with varying levels of imbalance ($\alpha \in [0.1, 50]$). A larger $\alpha$ indicates a more balanced feature distribution.
  - **Findings**: Both FeT and baseline models show improved performance in more balanced scenarios. FeT consistently demonstrates better accuracy than baselines across varying levels of heterogeneity.

- **Table 11**: Comparison of training efficiency on an RTX3090 GPU (batch size 128). PE: positional encoding; DM: dynamic masking.
  - **Findings**:
    1. **Parameter Efficiency**: FeT has a comparable or even smaller number of parameters than FedSim, suggesting that its performance improvement is due to model design rather than an increase in parameters.
    2. **Memory Efficiency**: FeT is significantly more memory efficient than FedSim, consuming only 20-39% of the memory. However, this efficiency comes at the cost of training speed, as FeT performs neighbor search in parallel during real-time training. In contrast, FedSim requires hours to link top-K neighbors and preloads all linked neighbors into GPU memory, resulting in longer linkage times and higher memory usage.
    3. **Overhead of New Components**: The additional components in FeT - dynamic masking and positional encoding - add very few extra parameters (1k - 0.4M), resulting in negligible additional computational cost (0-5 seconds per epoch slowdown).

- **Figure 14**: Impact of varying the number of neighbors ($K$) on FeT performance.
  - **Findings**: FeT consistently outperforms all baselines, even when $K > 50$, which involves many unrelated data records in the linkage process. When $K$ is small, the performace of FeT may decrease for the lack of infomation, the similar reason for the low performanae of Top1Sim. This demonstrates the ablity of FeT to filter out unrelated records in large $K$ scenarios.

- **Figure 15**: Real-world application of fuzzy multi-party VFL for travel cost prediction in a city.
  - **Application**: This scenario involves predicting travel costs through collaboration among companies managing taxis, cars, bikes, and buses. Since personal travel information is private and cannot be shared across companies, the identifiers for each route - starting and ending GPS locations - can only be linked fuzzily. Nevertheless, linking routes with closely related source and destination points significantly improves prediction accuracy.

In the detailed response to each reviewer, we use abbreviation to refer to each comment due to the charater contraint. Specifically,
- Wi: Weakness i
- MWi: Minor Weakness i
- Qi: Question i
- Li: Limitation i

Some concerns in the review are tightly related (e.g., Weakness 1 and Limitation 6), thus we have combined them (e.g., W1, L6) in our response.

In our response to each reviewer, we believe we have addressed all concerns and would be grateful if you could consider adjusting your rating if you find our revisions satisfactory.

---

### Author Response · Authors · 2024-08-10
**Summary of Rebuttal at the Three-Day Deadline for Author-Reviewer Discussion**

We understand the time and effort required to review our rebuttal. To assist you during the discussion phase, we have provided a brief summary of the major concerns and our responses for your convenience. With only **three days** remaining, we sincerely appreciate your consideration of our response and kindly request that you let us know if you have any further comments.

### Pros
- Reviewers `jPVu`, `sPbx`, and `so53` recognize the novelty of our framework design.
- Reviewers `jPVu`, `sPbx`, and `P7Jg` acknowledge our significant performance improvements over existing approaches.
- Reviewer `sPbx` appreciates the broader impact of this paper on multimodal learning.
- Reviewer `jPVu` highlights the effectiveness of our proposed privacy mechanism—SplitAvg—in reducing noise while preserving the same level of privacy.

### Major Concerns and Our Response

**Efficiency of FeT**: We compared FeT's computational and memory efficiency with FedSim, the state-of-the-art fuzzy VFL model, as shown in Table 11 of "rebuttal.pdf". We find that FeT delivers **better performance and improved memory efficiency with a similar number of parameters** compared to FedSim, despite a slightly lower training speed.

**FeT's Reliance on Linkage Quality**: FeT is **significantly less reliant** on initial linkage quality than existing VFL models, as demonstrated by three pieces of evidence in "rebuttal.pdf":
- **Ablation Study on K** (Figure 14): FeT consistently outperforms baselines even when many unrelated data records are included (K > 50), indicating resilience to low linkage quality.
- **Performance with Different Fuzziness** (Figure 12): FeT outperforms baselines under conditions of moderate to high noise in identifiers, showcasing strong resilience to fuzziness.
- **Dynamic Masking Visualization** (Figure 11): The visualization demonstrates that dynamic masking plays a crucial role in handling low-quality linkage, effectively focusing on a few close and relevant records among 4,900 fuzzily linked ones.

**Performance of FeT in "Less Ideal" Cases**: FeT performs better in **general fuzzy VFL scenarios** but slightly underperforms Top1Sim [4,5] in **ideal cases with precisely matched identifiers**, as evidenced by Figure 12 in "rebuttal.pdf" and Appendix D.

**Real Application of Multi-party Fuzzy VFL**: In Figure 15 of "rebuttal.pdf," we present a real-world application involving travel cost prediction in a city, achieved through collaboration among taxi, car, bike, and bus companies.

**Design of Dynamic Masking**: We used an ablation study (Table 3 of Appendix C.1) to show that **dynamic masking is necessary**; without it, accuracy may drop by up to 13%. Additionally, a visualization in Figure 11 of "rebuttal.pdf" shows that **dynamic masking learns to focus on a sample-specific localized area around the primary party's identifiers**.

**VFL with Limited Resources**:
1. **FeT's Limited GPU Memory Performance**: Table 11 of "rebuttal.pdf" shows that FeT performs well even with GPU memory under 1 GB, making it viable for resource-constrained VFL scenarios.
2. **Cross-Silo VFL Prevalence**: Cross-device VFL with limited resources is rare. VFL typically occurs in cross-silo settings [3], involving dozens to hundreds of parties with adequate computational power. Discussions with industry collaborators providing commercial VFL services confirm that it is uncommon for a single user to have features spread across thousands of distinct parties.

**Prevalence of Multi-party Fuzzy VFL in the Real World**: Traditional VFL assumes that each data record represents a **user**, making the assumption of a universal user ID natural. However, in real-world applications, these records often represent **various real-world objects**—such as a room (e.g., `house` and `hdb` datasets) or a travel route (e.g., `taxi` dataset).

Reviewers have also provided many useful comments on topics such as privacy, definitions, heterogeneous features, etc. We have addressed all these concerns in detail in our response to each reviewer. We greatly appreciate the reviewers' time and efforts in helping us improve this manuscript and are eager to know if you have any further comments regarding our response. Given the three-day constraint, your early reply would be greatly appreciated.

---

### Decision · Program_Chairs · 2024-09-25

**Decision:**

Accept (poster)

**Comment:**

This paper tackles the problem of multi-party VFL for fuzzily linked data (i.e., where the matching of samples across VFL clients is imperfect).
While multi-party or fuzzy VFL have been researched in prior work, the combination of the two has not.
The paper argues the multi-party problem introduces unique challenges such as more complex fuzzy linkage and higher privacy loss and seeks to mitigate these issues through several strategies including positional encoding averaging, dynamic masking of incorrect pairs, and SplitAvg that combines encryption and noise-based approaches to privacy.

The reviewers were generally positive about the paper after better understanding the problem and motivation. Clarification regarding the practical applicability and context for multi-party fuzzily linked data was noted by multiple reviewers and should be included in the final paper. Additionally, the authors should change the title to include "multi-party" as they suggested and also incorporate this into the paper more carefully to avoid overclaiming their contributions.

While there were some concerns regarding computational issues or experimental conditions, the real-world VFL datasets and author response reasonably addressed these concerns. I would recommend incorporating these discussions into the final paper.

In summary, this paper provides an interesting and effective solution to the multi-party fuzzy linked problem using a distributed transformer architecture that has practical real-world relevance for VFL scenarios.